

# Fermionic duality: General symmetry of open systems with strong dissipation and memory

**Valentin Bruch[1,2], Konstantin Nestmann[1,2],**
**Jens Schulenborg[3] and Maarten R. Wegewijs[1,2,4]**

**1** Institute for Theory of Statistical Physics, RWTH Aachen, 52056 Aachen, Germany
**2** JARA-FIT, 52056 Aachen, Germany
**3** Center for Quantum Devices, Niels Bohr Institute,
University of Copenhagen, 2100 Copenhagen, Denmark
**4** Peter Grünberg Institut, Forschungszentrum Jülich, 52425 Jülich, Germany

## Abstract

We consider the exact time-evolution of a broad class of fermionic open quantum systems with both strong interactions and strong coupling to wide-band reservoirs. We present a nontrivial fermionic duality relation between the evolution of states (Schrödinger) and of observables (Heisenberg). We show how this highly nonintuitive relation can be understood and exploited in analytical calculations within all canonical approaches to quantum dynamics, covering Kraus measurement operators, the Choi-Jamiołkowski state, time-convolution and convolutionless quantum master equations and generalized Lindblad jump operators. We discuss the insights this offers into the divisibility and causal structure of the dynamics and the application to nonperturbative Markov approximations and their initial-slip corrections. Our results underscore that predictions for fermionic models are already fixed by fundamental principles to a much greater extent than previously thought.

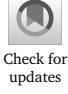

# 1  Introduction

The dynamics of open quantum systems is a problem of interest in a range of research fields. Their higher complexity as compared to closed systems evolving unitarily continues to motivate the development of new frameworks and approximation schemes to make further progress. Complementary to this, it has become more important to maximally reduce this complexity within existing well-developed approaches using basic symmetries and other general structures, see, e.g., Ref. [1] and references therein. For closed quantum systems, simplification by exploiting symmetries for some *fixed* set of system parameters is a highly developed subject and builds on the unitarity of transformations and the corresponding Hermicity of its generators. When turning to dynamics of open systems one runs into interesting new problems because the latter properties are lost in a reduced description.

In this paper we instead consider a different kind of simplification offered by a *duality mapping* in which the dynamics of a *fermionic* open system of interest is associated in a simple way to the dynamics of a similar system *governed by different parameters*.

**What is fermionic duality?**   The idea of the duality mapping is particularly easy to describe for a closed quantum system evolving unitarily with a time-constant Hamiltonian $H$. In this case the mapping explicitly constructs the adjoint Heisenberg evolution (superscript H) from the Schrödinger one by a substitution of physical parameters:

$$U^{\mathrm{H}}(t) := e^{iH^{\mathrm{H}}t} = U(t)^{\dagger} = e^{iHt} = e^{-i\bar{H}t} =: \bar{U}(t). \tag{1}$$

We will denote such a parameter mapping by an overbar. In the present simple example of a duality, the required relation between the Hamiltonian evolution generators,

$$H^{\mathrm{H}} = H = -\bar{H}, \tag{2}$$

is achieved by inverting the signs of all energies $H \to -H$: all local energies, all hopping amplitudes and all many-body interactions. To motivate this duality mapping consider the computation of the evolution of an arbitrary state, $|\psi(t)\rangle = \sum_i |u_i\rangle u_i(t)\langle u_i|\psi(0)\rangle$ which requires both the right eigenvectors $\{|u_i\rangle\}$ of $U(t)$ and its left eigenvectors $\{\langle u_i|\}$. Equivalently, one needs the right eigenvectors of $U(t)$ and of $U^{\mathrm{H}}(t) = \bar{U}(t)$ where we consider the Heisenberg evolution "as" a Schrödinger evolution at different parameter values. The duality mapping (1) makes explicit that these two sets of eigenvectors are related in a simple way through their *parameter dependence* allowing unnecessary algebra to be bypassed.

Having outlined the key idea, we immediately observe that for closed systems with time-constant $H$ this trick is completely pointless because there is an obvious shortcut: the eigenvectors of $U(t)$ are time-constant and coincide with the eigenvectors of $H = H^{\dagger}$ which are related by taking the Hermitian adjoint, $\langle u_i| = \langle h_i| = [|h_i\rangle]^{\dagger} = [|u_i\rangle]^{\dagger}$. Since $[H, U(t)] = 0$, time-dependence arises only through the eigenvalues $u_i(t) = e^{-ih_i t}$ where $h_i$ are the constant energy eigenvalues. Also, one may hesitate to work with the Hamiltonian $\bar{H}$ since it is clearly *unphysical*: inverting energies destabilizes any physical system which does not have an upper bound on its energy spectrum. Notably, for a *fermionic* system with a finite number of modes this latter objection is not really an issue since its spectrum is bounded by the Pauli exclusion principle.

However, when considering an open system with evolving density operator $\rho(t) = \Pi(t)\rho(0)$ the above mentioned shortcut completely breaks down. Although it turns out that non-unitary open-system evolutions can still be generated time-locally [2–5] as $\partial_t \rho(t) = -i\mathcal{G}(t)\rho(t)$, new problems arise because the generator is a *time-dependent, non-Hermitian* superoperator $\mathcal{G}(t) \neq [\mathcal{G}(t)]^{\dagger}$ even though the total system evolution is generated by a time-constant, Hermitian Hamiltonian operator. Physically, these new complications derive from memory (retardation) and dissipation effects, hallmarks of open-system dynamics. They cause the left and right eigenvectors of the generator $\mathcal{G}(t)$ to be distinct, time-dependent and different from the eigenvectors of the evolution propagator $\Pi(t)$ which is ultimately of interest: $[\mathcal{G}(t), \Pi(t)] \neq 0$. This implies that for an open system the transformation between the Schrödinger and Heisenberg generators is highly nontrivial (Ref. [4], p. 125) unlike the relation (2) for the underlying closed total system. An alternative simple mapping between the Schrödinger and Heisenberg picture evolution would dramatically simplify time-evolution calculations by providing a link between the left and right vectors.

Transposing the simple duality mapping for fermions from a closed to an open system does not seem to be possible at first: it is unclear how to evaluate the average of the simple relation (1) or (2) over the reservoir degrees of freedom (partial trace), even when making specific microscopic model assumptions. This is in contrast to other closed-system duality mappings [6–8] which are distinct from the one considered here [9]. It is therefore remarkable that for a very large class of fermionic open systems there does exist a nontrivial and useful extension of the duality which applies to the reduced time-evolution superoperator $\Pi(t)$. Anticipating its later detailed discussion [Eq. (23)], it provides an elegant formula for the adjoint

Heisenberg evolution analogous to Eq. (1) [10]:

$$\Pi^{\mathrm{H}}(t) := \left[\Pi(t)\right]^{\dagger} = e^{-\Gamma t}\mathcal{P}\,\bar{\Pi}(t)\,\mathcal{P}\,. \tag{3}$$

Here $\mathcal{P}$ is a linear transformation involving the fermion parity. Its presence hints at fermion parity superselection—forbidding quantum superpositions of states with even and odd fermion parity—as one fundamental principle on which the duality (3) is based [10]. $\Gamma$ is the lump sum of *microscopic* tunneling *constants*—known by inspecting the underlying model Hamiltonian—and the overbar again denotes a *parameter mapping*. This generalization of Eq. (1) is truly dissipative. For example, for a resonant level coupled to a reservoir the parameter mapping inverts not only the *sign* of the level energy $\varepsilon$ and the electrochemical potential $\mu$, but also the dissipative *tunnel-rate* constant $\Gamma$. The relation (3) was first derived in Ref. [10] without making weak coupling and / or "Markovian" assumptions, requiring the techniques of Refs. [11–14] to explicitly consider all orders of the expansion of $\Pi(t)$ in the system-environment coupling. In the following we will denote this direct consideration of the exact propagator $\Pi(t)$ as approach (i) to duality.

Applied to weakly coupled but locally interacting open systems, the fermionic duality (3) has already provided several interesting insights and predictions [10,15–17]. For example, the time-dependent response of a "kicked" quantum dot with *repulsive* Coulomb interaction was shown to exhibit effects of electron-*attraction*. This surprising effect can be nicely understood from the duality mapping which involves the inversion of the local interaction parameter as in Eq. (2). This explains pronounced effects in the measurable time-dependent heat current which is sensitive to interactions. The same formulas are very difficult to understand directly in terms of the real repulsive interaction, but are easily rationalized by electron-pairing induced by the attraction in the fictitious dual system defined by the duality mapping. More generally, the thermoelectric response of a quantum dot—although studied long ago—entails several features that turned out to have a very simple explanation in terms of an effective attractive model that is dual to the repulsive system of interest [15, 16]. These conclusions hold even beyond linear response to electro-thermal biases where, e.g., Onsager relations no longer apply, and the effects can be understood by extending the weak-coupling fermionic duality beyond the wide-band limit [16]. In all these cases, the original system is analyzed by a dual system, an *effective* system with at worst unconventional properties. Conversely, it was also shown that the response of a physically attractive dual system can be understood better by exploiting its repulsive original system [17]. Thus in the weak-coupling limit the duality can also be used in reverse.

**Extension to other approaches.** So far, these applications were in fact all based on a different formulation of the duality which we will denote by approach (ii) in the following. It differs from Eq. (3) by relying on the time-nonlocal quantum master equation (QME) also called Nakajima-Zwanzig (NZ) [18,19] or time-convolution type QME. By introducing a memory kernel it anticipates the time-convolution structure of the higher-order system-reservoir coupling terms encountered in the microscopic derivation of the duality [10]. Whereas in the weak-coupling limit approach (ii) recovers various other types of quantum master equations, for the interesting regime of stronger coupling it differs in essential points. So far it has remained unclear what the implications of the fermionic duality are in general for other approaches to open-system dynamics. This problem is solved in the present paper: besides extending the propagator approach (i) and the memory kernel approach (ii), we establish the fermionic duality for three additional approaches which are fundamentally different and complementary as we now outline.

(iii) The Sudarshan-Kraus or *measurement-operator approach* [20–22]—ubiquitous in quantum-information theory—also directly addresses the time-evolution superoperator $\Pi(t)$.

However, it is an *operational* approach which decomposes the evolution into independent physical processes conditioned on possible outcomes of measurements performed on the system's environment. Theoretically, this has the distinct advantage that approximations formulated in terms of these operational building blocks automatically preserve the positivity of quantum states, also in the presence of initial entanglement with a reference system (complete positivity). From these measurement operators acting only on the system one can furthermore compute the evolution of its *effective environment* and quantify the exchange of information as illustrated in Ref. [23]. Barring special limits where simpler Lindblad equations [24, 25] apply (Markovian semigroups), for most systems of interest the insights offered by this approach seem practically impossible to gain in other formulations. The same applies to the so-called Choi-Jamiołkowski state, which is closely related to the measurement-operator sum by a well-known isomorphism [26–28]. The microscopic calculation of Kraus operators has received recent attention [29–33] but remains very difficult, motivating our search for analytic simplifications.

(iv) The time-convolutionless (TCL) or *time-local quantum master equation approach* [2, 3, 34] has the advantage that it allows the Markovianity of the evolution to be scrutinized more conveniently through the time-local generator $\mathcal{G}(t)$ mentioned earlier. It is thus closely tied to the question of the divisibility of the dynamics [35, 36]. In practice, time-local QMEs also arise naturally from the *time-nonlocal* QMEs of approach (ii) when consistently accounting for frequency dependence of the memory kernel in decay problems [37, 38]—recently generalized in Ref. [5]—or in adiabatic expansions for situations with external driving [9, 39–41]. The microscopic calculation of $\mathcal{G}(t)$ is, however, very challenging making additional analytic insights valuable [34, 42, 43].

(v) The closely related *jump-operator approach* decomposes the time-local generator $\mathcal{G}(t)$ into "quantum jumps" with intermittent renormalized Hamiltonian evolution occurring at infinitesimal time steps. Although this is similar to the measurement-operator approach (iii) it provides distinct insights by primarily making the conditions for divisibility—rather than complete positivity—explicit. Importantly, this approach is also at the basis of the successful stochastic simulation method for open-system dynamics [44–51] and includes the familiar Markovian Lindblad QMEs as a special case. In the present work the jump-operator approach is particularly interesting because it most explicitly generalizes the closed-system duality (2) discussed above.

In none of the approaches (iii)–(v) the implications of fermionic duality have been explored. Doing so is of particular interest since these methods are pivotal for the continued fruitful application of ideas from quantum information theory to open-system dynamics [4, 50, 52–56]. One should note that although approaches (i)–(v) are exactly equivalent, they define completely different starting points for approximations and formal considerations. Thus, having a formulation of fermionic duality in hand for each case will enable attaining independent insights. This holds true even when applied to the simplest, explicitly solvable transport model of a strongly coupled resonant level as was recently highlighted in Ref. [23] and we will draw on this reference for illustration. Even though this model has been studied for decades [57] the fermionic duality relations presented here went unnoticed. Importantly, our results continue to hold for a large class of much more complicated models whose detailed discussion is however beyond the present scope.

**Fermionic duality: Useful but unphysical?** Before proceeding it is important to neutralize two potential points of confusion. An immediate worry is that the fermionic duality for open systems maps some physical parameters to *unphysical* values as noted above. In fact, for open systems the unphysical destabilization of the system by inverting the signs of all local energies discussed after Eq. (1) becomes more prominent. For one, the duality mapping even

makes the system-reservoir *coupling* Hamiltonian *anti*-Hermitian, $H_T \to iH_T$. Although no real physical parameters become imaginary, this does *invert the sign of all dissipative decay rates*. As mentioned, for a resonant level this means that the decay constant is inverted $\Gamma \to -\Gamma$. This does not lead to divergent quantities since in the duality relation (3) the negative decay rates are explicitly compensated by an exponentially decaying prefactor $e^{-\Gamma t}$. Moreover, in the weak coupling limit close inspection reveals [10] that one can use fermionic duality to set up a relation between two dual *physical* systems which both have nonnegative decay constants but otherwise different physical parameters. This simplification facilitated the applications in the weak coupling limit cited earlier. In the present paper we will, however, focus on the general case of strong coupling where this simplification fails[1] and this non-physicality must be confronted.

We will show that the anti-Hermitian coupling Hamiltonian causes the *reduced* dynamics $\bar{\Pi}(t)$ to violate complete positivity, giving a clear operational meaning to the vague notion of an "unphysical" system. This is important since it will allow us to identify which contributions to the evolution of the dual system are unavoidably unphysical, a question that cannot be answered directly using the original derivation of the duality in Ref. [10]. Instead, by leaving aside the derivation and only considering the duality relation (3) as such, this paper shows that the loss of complete positivity is associated with the fermionic parity transformation $\mathcal{P}$, a key ingredient of the duality. Hence, unlike in the weak coupling limit, the dual evolution has no statistical meaning anymore and one can no longer refer to an effective, *physical* dual system which simulates the original system. Although this may sound disastrous at first, it will become clear that in none of the discussed approaches these unintuitive features of fermionic duality limit its practical usefulness. Since in each of these approaches the fundamental property of complete positivity is expressed—if at all—by different constraints, a careful discussion what is unphysical about the dual equations will be a recurring side-theme. It will emerge that the general unphysicality of fermionic duality is instead of an artifact a key feature unveiling its unconventional insights as compared to ordinary symmetries, see Sec. 6.

To avoid confusion about the domain of applicability we note that the fermionic duality (3) is primarily important for *analytical* calculations where one obtains some quantities of interest as *functions* of physical parameters. By a simple substitution of parameters it allows one to bypass very complicated and nonintuitive algebra. The ultimate importance of fermionic duality lies therein that this simplification allows the analysis of physical effects [10, 15–17] to be pushed much further. Unlike doing algebra, solution by parameter substitution has the advantage that it preserves the compact form of an expression that *has already been calculated, simplified and physically* well-understood. For example, it makes explicit which quantities have a similar functional dependence: if one knows that some contribution is an exponential function of time then generically the dual contribution obtained by a parameter substitution is exponential as well. Loosely speaking, one can thus distinguish *individual* nontrivial contributions to the dynamics (non-exponential time dependence) from trivial (exponential) ones. The duality mapping also implies the concept of self-dual quantities: Despite being physical, such quantities are mapped onto themselves by the generally unphysical duality relation, and are thereby constrained in ways that are impossible to see with common physical dualities, symmetries or intuition.

**Outline.** The outline of the paper is as follows. In Sec. 2 we first consider the simplest, exactly solvable open system that exhibits fermionic duality beyond the weak-coupling limit [16], the resonant level. This provides the simplest yet nontrivial illustration of the general results derived in the subsequent sections. In Sec. 3 we consider the fermionic duality in its most basic form (3) obtained in Ref. [10] as a mapping between the *finite-time* Schrödinger and

---

[1]see footnote 20 at Eq. (70).

Table 1: Duality relations for the approaches numbered (i)–(v) mentioned in the introduction. By $\stackrel{!}{=}$ we indicate equalities that are valid only in the special case where the evolution commutes with its generator, $[\mathcal{G}(t), \Pi(t)] = 0$, which includes the weak-coupling limit where $\Pi(t) = e^{-i\mathcal{G}t}$ with constant $\mathcal{G}$. Hat denotes the Laplace transform $\hat{f}(\omega) = \int_0^\infty dt\, e^{i\omega t} f(t)$. Bar denotes the duality mapping of parameters which effects $H \to -H$, $H_\mathrm{T} \to iH_\mathrm{T}$ and $\mu_r \to -\mu_r$. $\mathcal{I}$ is the identity superoperator.

|  | Finite evolution approaches | Infinitesimal evolution approaches |
|---|---|---|
| **Super-operator approaches** | (i) Propagator $\Pi(t)$ [Sec. 3.1] $$\Pi(t)^\dagger = \Pi^\mathrm{H}(t)$$ $$= e^{-\Gamma t}\mathcal{P}\bar{\Pi}(t)\mathcal{P}$$ $$\hat{\Pi}(-\omega^*)^\dagger = \widehat{\Pi^\mathrm{H}}(\omega)$$ $$= \mathcal{P}\widehat{\bar{\Pi}}(\omega + i\Gamma)\mathcal{P}$$ | (ii) Time-nonlocal memory kernel [Sec. 4.3] $$\left[\mathcal{K}(t)\right]^\dagger = \mathcal{K}^\mathrm{H}(t)$$ $$= i\Gamma\mathcal{I}\delta(t) - e^{-\Gamma t}\mathcal{P}\bar{\mathcal{K}}(t)\mathcal{P}$$ $$\hat{\mathcal{K}}(-\omega^*)^\dagger = \widehat{\mathcal{K}^\mathrm{H}}(\omega) = i\Gamma - \mathcal{P}\widehat{\bar{\mathcal{K}}}(\omega + i\Gamma)\mathcal{P}$$ (iv) Time-local generator [Sec. 4.1] $$\mathcal{G}(t)^\dagger \stackrel{!}{=} \mathcal{G}^\mathrm{H}(t) = \left[\Pi(t)^{-1}\mathcal{G}(t)\Pi(t)\right]^\dagger$$ $$= i\Gamma\mathcal{I} - \mathcal{P}\bar{\mathcal{G}}(t)\mathcal{P}$$ |
| **Operational approaches** | (iii) Measurement operators [Sec. 3.2] $$M_\alpha(t)^\dagger = \bar{M}_{\alpha'}(t)$$ $$m_\alpha(t) = e^{-\Gamma t}(-1)^{N_{\alpha'}}\bar{m}_{\alpha'}(t)$$ | (v) Jump operators and effective Hamiltonian [Sec. 4.2] $$J_\alpha(t)^\dagger \stackrel{!}{=} J_\alpha^\mathrm{H}(t) = \bar{J}_{\alpha'}(t)$$ $$j_\alpha(t) \stackrel{!}{=} j_\alpha^\mathrm{H}(t) = (-1)^{N_{\alpha'}}\bar{j}_{\alpha'}(t)$$ $$H(t) \stackrel{!}{=} H^\mathrm{H}(t) = -\bar{H}(t)$$ |

Heisenberg superoperators. From this we derive a fermionic duality for the set of Kraus measurement operators using the Choi-Jamiołkowski state associated with the dynamics. In Sec. 4 we consider fermionic duality for the *infinitesimal-time generators* of the evolution, either via a time-local or time-nonlocal quantum master equation. Whereas the time-nonlocal formulation allows for a solution in the Laplace-frequency domain, the time-local formulation allows for a further decomposition into jump operators. This leads to some unexpected insights into the divisibility of the dynamics and its causal structure. Finally, in Sec. 5 we combine these approaches to gain deeper insight into a generally applicable nonperturbative semigroup approximation [5, 23] and its correction by an "initial slip". Here we combine the fermionic duality with a recently found exact functional relation between the *time-local* generator $\mathcal{G}$ and the *time-nonlocal* memory kernel $\mathcal{K}$ [5]. In Sec. 6 we conclude and outline directions for follow-up work. Table 1 provides a guide to the paper by summarizing the duality relations for all discussed approaches. Throughout the paper we set $\hbar = k_B = 1$.

## 2 Simple example: Fermionic resonant level

The general fermionic duality is best illustrated for the simple model of a resonant level with arbitrary tunnel coupling $\Gamma$ to a single reservoir at temperature $T$ and electrochemical potential $\mu$,

$$H_\mathrm{tot} = \varepsilon d^\dagger d + \int d\omega\, \omega b_\omega^\dagger b_\omega + \sqrt{\frac{\Gamma}{2\pi}}\int d\omega \left(d^\dagger b_\omega + b_\omega^\dagger d\right), \tag{4}$$

where $d$ and $b_\omega$ are fermionic annihilation operators. This gives only a very basic description of the time-dependent (dis)charging of quantum dot systems realized in a range of heterostructures including molecular junctions [58–60] and atomic impurities [61]. Because it ignores

Coulomb interaction effects this model is exactly solvable for strong coupling $\Gamma$ whose effects are primarily of interest here. Although the solution is known since Ref. [57], several interesting aspects of the dynamics of the density operator $\rho(t)$ were overlooked until recently [23], including measurable effects of the breakdown of two different notions of Markovianity. In the present paper we complement the study [23] by pointing out further interesting properties of this model which are implied by fermionic duality and which, importantly, turn out to be more generally valid. For example, the spectral properties of the various time-evolution quantities of approaches (i), (ii) and (iv) were noted in Ref. [23] to display striking regularities which could not be rationalized by any symmetry of the resonant level model. The measurement- and jump-operators of approach (iii) and (v) were likewise found to exhibit a pronounced pattern and to obey an unexpected sum rule whose form depends only on one microscopic parameter and time. These formal regularities indicate that there is still more to be said about the physical properties of this model, for instance, its degree of (non-)Markovianity as introduced below. As we will see in Sec. 5 this ties in with the practical task of constructing Markovian semigroup approximations to the solution and their corrections. All these points—and more—will be explored step by step in this paper.

*Physical properties of the dynamics.* To appreciate these implications of fermionic duality, we, however, first need to give a summary of the various analytical forms of the resonant level model dynamics for later reference. Further details on the latter are given in Ref. [23]. Explicitly, all nontrivial dependence on the level *detuning* $\varepsilon - \mu$, temperature $T$ and tunnel coupling $\Gamma$ is captured by three related functions of time:

$$\mathsf{k}(t) = 2T \, \frac{\sin((\varepsilon - \mu)t)}{\sinh(\pi T t)}, \tag{5}$$

$$\mathsf{g}(t) = \int_0^t ds \, e^{-\frac{1}{2}\Gamma s} \mathsf{k}(s), \tag{6}$$

$$\mathsf{p}(t) = \frac{\Gamma}{1 - e^{-\Gamma t}} \int_0^t ds \, e^{-\Gamma(t-s)} \mathsf{g}(s). \tag{7}$$

These functions contain pronounced damped oscillatory contributions at low $T$. Fortunately their complicated precise form given in Ref. [23] is not needed here.

We first review how the functions $\mathsf{k}$, $\mathsf{g}$ and $\mathsf{p}$ encode basic physical properties of the dynamics which will be important later on. For fixed time $t$, the state-evolution map $\rho(0) \mapsto \rho(t) = \Pi(t)\rho(0)$ has two fundamental properties, namely trace-preservation (TP), $\mathrm{Tr}\,\Pi(t)\rho(0) = \mathrm{Tr}\,\rho(0)$ for all initial states $\rho(0)$, and complete positivity (CP)—reviewed in App. A—which implies $\Pi(t)\rho(0) \geq 0$ for all $\rho(0) \geq 0$. Although TP imposes no restrictions on $\mathsf{k}$, $\mathsf{g}$ and $\mathsf{p}$, the CP property is fully encoded in the range of values that the function $\mathsf{p}(t)$ can take at any time: $\Pi(t)$ is CP if and only if $|\mathsf{p}(t)| \leq 1$. This nontrivial constraint is indeed [23] satisfied by formula (7) for all times $t$ and *all physical values* of the parameters $\varepsilon, \mu, T, \Gamma$ of the model, as it should. Depending on these parameters, the dynamics may be Markovian in two different ways with different observable consequences as we now explain.

(1) The dynamics may be divisible *by itself*, $\Pi(t) = \Pi(t - t')\Pi(t')$, for all $t \geq t'$, which is the case if and only if the function $\mathsf{g}(t) = $ constant for $t > 0$. Despite the simplicity of the model, this *semigroup*-divisibility always breaks down except at resonance, $|\varepsilon - \mu|/T \to 0$, where $\mathsf{p}(t) = 0$ (which always holds in the limit of a hot reservoir, $T \to \infty$), or, when the level is completely off-resonance, $|\varepsilon - \mu|/T \to \infty$, where $\mathsf{p}(t) = \pm\Theta(t)$ is a step function. Thus, the dynamics is virtually never Markovian in the semigroup sense and cannot be described by a Lindblad quantum master equation. The breakdown of this property is witnessed by an anomalous transient *enhanced* level occupation when decaying to a *more depleted* stationary state [23].

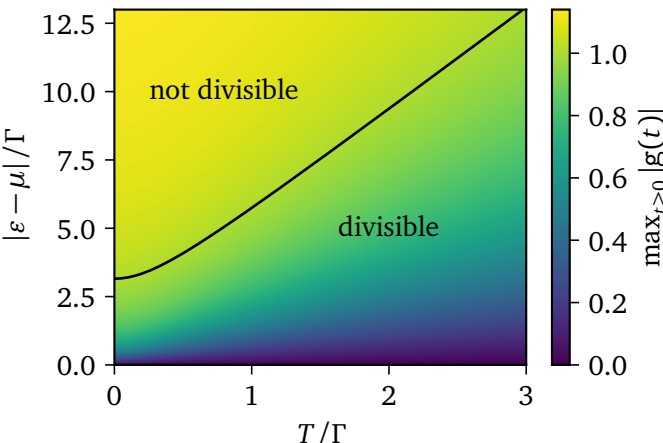

Figure 1: CP-divisibility for the resonant level model. The maximal value $\max_{t\geq 0}|\mathsf{g}(t)|$ is plotted as function of level detuning $\varepsilon - \mu$ and temperature $T$ in units of tunnel coupling $\Gamma$. Whenever $|\mathsf{g}(t)| > 1$ for some time $t$ the dynamics is not CP-divisible. The black curve marks where the maximal value equals 1. Note that $\max_{t\geq 0}|\mathsf{g}(t)|$ depends only on the ratios $(\varepsilon - \mu)/\Gamma$ and $T/\Gamma$, whereas $\mathsf{g}(t)$ like $\mathsf{k}(t)$ depends on all parameters separately.

(2) In a less strict sense, the dynamics may still be divisible as $\Pi(t) = \Pi(t, t')\Pi(t')$ for all $t \geq t'$ by *another physical* evolution $\Pi(t, t')$, a CP-TP map [35, 36]. This *CP-divisibility* turns out to occur if and only if $|\mathsf{g}(t)| \leq 1$ for all $t$. This nontrivial condition is mapped out in Fig. 1 as function of level position and temperature relative to the coupling energy. One sees that the dynamics fails to be Markovian in the sense of CP-divisibility whenever the level is off-resonant by more than the tunneling *and* thermal broadening, $|\varepsilon - \mu| \gtrsim \max\{\Gamma, T\}$. For the resonant level this distinct property can be observed in transport by checking whether there is no reversal of the measured current as function of time for *any* initial level occupation [23].

*Evolution.* Having outlined some of the physics of this model, we now describe how the functions (5)–(7) explicitly determine the structure of the exact dynamics of $\rho(t)$. The evolution can be written in three different ways and features the function $\mathsf{p}(t)$ [Eq. (7)]:

$$\Pi(t) = \exp\left(-i[H, \bullet]t + \tfrac{1}{2}\Gamma t \sum_{\eta = \pm}[1 - \eta\mathsf{p}(t)]\mathcal{D}_\eta\right) \tag{8}$$

$$= \sum_{i=0}^{3} \pi_i(t)\big|\pi_i(t)\big)\big(\pi_i'(t)\big| \tag{9}$$

$$= \sum_{N=0,1}\sum_{\eta=\pm} m_{N\eta}(t)M_{N\eta}(t)\bullet M_{N\eta}(t)^\dagger. \tag{10}$$

By $[H, \bullet]$ we denote the commutator of the system Hamiltonian[2] $H = \varepsilon d^\dagger d$ with argument $\bullet$, and the dissipator $\mathcal{D}_\eta$ will be defined in Eq. (11). Also, we write $\big(A\big|\bullet := \text{Tr} A^\dagger \bullet$ and $\big|B\big) := B$ for operators $A, B$. Throughout the paper we will label each right eigenvector by its eigenvalue and the corresponding left eigenvector is distinguished by an additional prime as in Eq. (9).

The exponential form (8) is particular to this simple model. Due to the nontrivial dependence of $\mathsf{p}(t)$ on time [Eq. (6)] it is *not* the exponential solution of some Lindblad equation,

---

[2]Throughout the paper we consider the action of the map $\Pi(t)$ on arbitrary initial states $\rho(0)$ since this enables the techniques of Refs. [11, 13, 14] which lead to the neat exponential form (8), see Ref. [23]. Also, this allows us later on [Eq. (61)] in the jump-operator approach (v) to generalize Eq. (2) of the introduction. If one restricts the action of the map $\Pi(t)$ to operators $\rho(0)$ which are fermionic states commuting with the parity, $[\rho(0), (-\mathbb{1})^N] = 0$ (superselection), then the contribution of the system Hamiltonian in Eq. (8) is not relevant *in this model*. For this restricted map one can also find a simpler set of measurement operators.

despite the appearance of the familiar dissipator superoperators

$$\mathcal{D}_\eta := d_\eta \bullet d_\eta^\dagger - \tfrac{1}{2}\{d_\eta^\dagger d_\eta, \bullet\}, \qquad \eta = \pm, \tag{11}$$

where we defined $d_+ := d^\dagger$, $d_- := d$. The more general spectral decomposition (9) is natural to approach (i). The eigenvalues and their distinct left and right eigenvectors for this model are listed in Table 2. Finally, the Kraus operator sum (10) of approach (iii) always exists and in the present case the measurement operators read

$$M_{0\eta}(t) = \eta\sqrt{v_\eta(t)}e^{i\frac{1}{2}\varepsilon t}dd^\dagger + \sqrt{v_{-\eta}(t)}e^{-i\frac{1}{2}\varepsilon t}d^\dagger d, \qquad M_{1\eta} = d_\eta, \tag{12}$$

with nonnegative coefficients

$$m_{0\eta}(t) = e^{-\frac{1}{2}\Gamma t}\left[\cosh\left(\tfrac{1}{2}\Gamma t\right) + \eta\sqrt{1 + \mathsf{p}(t)^2 \sinh\left(\tfrac{1}{2}\Gamma t\right)^2}\right], \tag{13a}$$

$$m_{1\eta}(t) = \tfrac{1}{2}\left(1 - e^{-\Gamma t}\right)[1 - \eta\mathsf{p}(t)], \tag{13b}$$

and the shorthand

$$v_\eta(t) = \frac{1}{2} + \frac{\eta}{2}\frac{\mathsf{p}(t)\sinh(\tfrac{1}{2}\Gamma t)}{\sqrt{1 + \mathsf{p}(t)^2 \sinh(\tfrac{1}{2}\Gamma t)^2}}. \tag{14}$$

*Quantum master equations.* The above dynamics is the solution of the exact *time-nonlocal* QME of approach (ii),

$$\frac{d}{dt}\rho(t) = -i\int_0^t dt'\,\mathcal{K}(t - t')\rho(t'), \tag{15}$$

whose memory kernel features the function $\mathsf{k}(t)$ [Eq. (5)],

$$-i\mathcal{K}(t) = -i[H, \bullet]\delta(t) + \tfrac{1}{2}\Gamma\sum_\eta\left[\delta(t) - \eta e^{-\frac{1}{2}\Gamma t}\mathsf{k}(t)\right]\mathcal{D}_\eta. \tag{16}$$

Note that we included the system Hamiltonian $H$ into $\mathcal{K}$ and used the normalization $\int_0^t ds\,\delta(s) = 1$. Finally, the dynamics is also the solution of an exact *time-local* QME that defines approach (iv),

$$\frac{d}{dt}\rho(t) = -i\mathcal{G}(t)\rho(t), \tag{17}$$

with a generator that features the function $\mathsf{g}(t)$ [Eq. (6)],

$$-i\mathcal{G}(t) = -i[H, \bullet] + \tfrac{1}{2}\Gamma\sum_\eta[1 - \eta\mathsf{g}(t)]\mathcal{D}_\eta \tag{18a}$$

$$= -i\sum_i g_i(t)\big|g_i(t)\big)\big(g_i'(t)\big|. \tag{18b}$$

The eigenvalues and eigenvectors in Eq. (18b) are listed in Table 2. Although Eq. (16) and Eq. (18a) look similar to the exponent of Eq. (8), they involve the three very different functions (5)–(7).

Having summarized the exact equations for this model to be discussed, we note that in the weak coupling limit it is not difficult to reveal a simple structure. For example, in the time-local QME (17) one can simplify $1 - \eta\mathsf{g}(t) \approx f(\eta(\varepsilon - \mu)/T)$ where $f$ is the Fermi function and one then directly derives a fermionic duality relation using the formal replacement $f((\varepsilon - \mu)/T) \to 1 - f((\varepsilon - \mu)/T)$ as explained in Ref. [16]. However, this is not possible in the case of arbitrarily strong coupling $\Gamma$ considered here. Thus, despite the simplicity of this solvable model none of the above representations of its exact dynamics seems to exhibit an obvious general structure. In the following we will derive such a structure and illustrate it for each of the above expressions.

Table 2: Time-dependent eigenvectors and eigenvalues of the evolution $\Pi(t)$ and its time-local generator $\mathcal{G}(t)$ derived in Ref. [23]. Observe that the eigenvalues are trivial exponentially decaying functions and constants, respectively, fixed by the $T = \infty$ semigroup limit of the model. All nontrivial time-dependence is due to finite-$T$ effects [23] and enters the dynamics through the functions $\mathsf{p}(t)$ and $\mathsf{g}(t)$ which appear only in the eigen*vectors* of $\Pi(t)$ and $\mathcal{G}(t)$, respectively. Note that here the normalization of the eigenvectors of $\mathcal{G}(t)$ differs from the normalization fixed in Eqs. (50b) and (50c). The duality relation for the coherences can be seen from $\mathcal{P}|d_\eta) = -\eta|d_\eta)$.

| | Spectral decomposition of $\Pi(t)$ | | |
|---|---|---|---|
| $i$ | $\left(\pi_i'(t)\right|$ | $\pi_i(t)$ | $\left|\pi_i(t)\right)$ |
| 0 | $(\mathbb{1}|$ | $1$ | $\frac{1}{2}\left[\left|\mathbb{1}\right) + \mathsf{p}(t)\left|(-\mathbb{1})^N\right)\right]$ |
| $1,2$ | $\left(d_\eta^\dagger\right|$ | $e^{(i\eta\varepsilon - \frac{1}{2}\Gamma)t}$ | $\left|d_\eta^\dagger\right)$ |
| 3 | $\frac{1}{2}\left[\left((-\mathbb{1})^N\right| - \mathsf{p}(t)\left(\mathbb{1}\right|\right]$ | $e^{-\Gamma t}$ | $\left|(-\mathbb{1})^N\right)$ |

| | Spectral decomposition of $\mathcal{G}(t)$ | | |
|---|---|---|---|
| $i$ | $\left(g_i'(t)\right|$ | $g_i(t)$ | $\left|g_i(t)\right)$ |
| 0 | $(\mathbb{1}|$ | $0$ | $\frac{1}{2}\left[\left|\mathbb{1}\right) + \mathsf{g}(t)\left|(-\mathbb{1})^N\right)\right]$ |
| $1,2$ | $\left(d_\eta^\dagger\right|$ | $-\eta\varepsilon - i\frac{1}{2}\Gamma$ | $\left|d_\eta^\dagger\right)$ |
| 3 | $\frac{1}{2}\left[\left((-\mathbb{1})^N\right| - \mathsf{g}(t)\left(\mathbb{1}\right|\right]$ | $-i\Gamma$ | $\left|(-\mathbb{1})^N\right)$ |

## 3 Fermionic duality for exact time-evolution

We now extend the scope to the much broader class of models of the form $H_{\text{tot}} = H + H_{\text{R}} + H_{\text{T}}$ where only the following assumptions are made: (I) The multiple fermionic reservoirs described by $H_{\text{R}}$ are noninteracting with structureless, infinitely wide bands, each one being separately in equilibrium at the initial time. (II) The coupling to the fermions in the system (indexed by $l$) is bilinear in the field operators, $H_{\text{T}} = \sum_{rl} \int d\omega \, t_{rl} \, d_l^\dagger c_{r\omega} + \text{h.c.}$, and independent of the energy $\omega$ of the fermionic modes in the reservoirs (indexed further by $r$). (III) The system Hamiltonian $H$ obeys parity superselection, $[H, (-\mathbb{1})^N] = 0$, and as a result so does the total system. The only microscopic quantity that explicitly plays a role in the fermionic duality is the lump sum of tunnel-coupling constants over the system and reservoir indices:

$$\Gamma := 2\pi \sum_{rl} |t_{rl}|^2. \tag{19}$$

Here $l$ ($r$) includes all relevant quantum numbers (spin, orbital moment, etc.) on the system (reservoir) which need not be conserved by $H_{\text{T}}$, unlike the fermion number.

Based on these three assumptions the duality was established in Ref. [10]. However, the detailed derivation given there does not lead to the insights reported in the present paper. The conditions (I)–(III) do not help to understand the results obtained here by starting from the established duality relation Eq. (3). We refer to Refs. [10, 16] for further discussion of these assumptions and the derivation and to Refs. [10, 13–17, 62] for numerous detailed illustrations of how the duality can be technically applied and physically exploited in the weak coupling limit.

No other assumptions are necessary: in particular, the system, described by $H$, may consist of *any* finite number of levels with *any* type of multi-particle interaction of arbitrary strength, including superconducting pairing terms that break particle conservation but preserve parity.

Also, the magnitude of the couplings, temperatures and electrochemical biases can be arbitrary assuming that the employed perturbation series converges. Thus, the following results apply to a very large class of actively studied models which are relevant to nonequilibrium quantum-impurity physics, quantum transport and open-system dynamics. We also note that for weak coupling, the duality relation can be generalized beyond the case of structureless wide bands [16].

Of central interest is the superoperator $\Pi(t)$ describing the state evolution, i.e., the Schrödinger propagator,

$$\rho(t) = \Pi(t)\rho(0) = \underset{R}{\text{Tr}} \left\{ e^{-iH_{\text{tot}}t} \rho(0)\rho_R e^{iH_{\text{tot}}t} \right\}. \tag{20}$$

It is obtained by tracing out the fermionic reservoirs, assuming that each of these is initially uncorrelated with the system and separately in an equilibrium state. The propagator is thus a function of the parameters specifying the system Hamiltonian $H$, the coupling $H_T$, and the different electrochemical potentials of the reservoirs, collected in $\mu = (\mu_r)$. This dependence is important in the following and will be denoted by $\Pi(t, H, \mu, H_T)$ when required. The dependence on the different reservoir temperatures $T_r$ need not be indicated.

The superoperator $\Pi(t)^\dagger$ describes the time-evolution of system observables $A$, i.e., the Heisenberg propagator,

$$A(t) = \Pi^H(t)A = \Pi(t)^\dagger A, \tag{21}$$

such that $\langle A(t)^\dagger \rangle_{\rho(0)} = (A(t)|\rho(0)) = (A|\rho(t)) = \langle A^\dagger \rangle_{\rho(t)}$ for expectation values. Here the superadjoint of a superoperator, indicated by bold $\dagger$, is defined by $\text{Tr}\left\{A^\dagger(\Pi B)\right\} = \text{Tr}\left\{(\Pi^\dagger A)^\dagger B\right\}$ and is of central importance in this paper. It is defined relative to the Hilbert-Schmidt scalar product between operators $(A|B) = \text{Tr}A^\dagger B$ and therefore distinct from the ordinary adjoint $\dagger$ of an operator relative to the scalar product between vectors $\langle \psi|A\phi \rangle = \langle A^\dagger \psi|\phi \rangle$. For superoperators with the special form $\bullet \mapsto (L \bullet R)$ of a left and right multiplication by operators $L$ and $R$, respectively, the two distinct adjoint operations are related in a simple way:

$$(L \bullet R)^\dagger = L^\dagger \bullet R^\dagger. \tag{22}$$

In the following these distinctions will be clear in the context and we will talk about adjoints, eigenvectors, and orthogonality without further specification ("super"). Since generally the evolution $\Pi(t)$ is a not represented by a normal matrix, $[\Pi(t)^\dagger, \Pi(t)] \neq 0$, its left and right eigenvectors are *not* simply related by taking the adjoint $\dagger$. As both sets of vectors are required in the analysis of dynamics, this presents a crucial complicating factor in any (semi-)analytical treatment of open quantum systems. This is what fermionic duality addresses.

## 3.1 Evolution superoperator

The fermionic duality establishes a relation between $\Pi^H(t) = \Pi(t)^\dagger$ and $\Pi(t)$ *evaluated at different parameter values* which is denoted by $\bar{\Pi}(t)$. By first explicitly evaluating the wideband limit, this relation can be derived within a renormalized perturbation expansion of all finite-$T$ corrections of the propagator $\Pi(t)$ around the $T \to \infty$ limit [10, 13, 14]. For the considered class of models the propagator obeys the *fermionic duality relation*

$$\Pi^H(t) = \left[\Pi(t)\right]^\dagger = e^{-\Gamma t}\mathcal{P}\bar{\Pi}(t)\mathcal{P} \tag{23}$$

order-by-order. Here $\Gamma$ is the lump sum of couplings (19) and the superoperator

$$\mathcal{P} = (-\mathbb{1})^N \bullet \tag{24}$$

denotes the left multiplication with the system parity operator $(-\mathbb{1})^N := \exp(i\pi N)$. By the overbar we denote the following parameter substitution of some function $X$:

$$\bar{X}(H, \mu, H_T) := X(-H, -\mu, iH_T). \tag{25}$$

For example, for the resonant level model of Sec. 2 this parameter mapping corresponds to $(\varepsilon, \mu, \Gamma) \rightarrow (-\varepsilon, -\mu, -\Gamma)$ which transforms the functions encoding all nontrivial parameter dependence as follows[3]:

$$\bar{\mathsf{k}}(t) = -\mathsf{k}(t), \tag{26}$$

$$\bar{\mathsf{g}}(t) = e^{\Gamma t}\big[-\mathsf{g}(t) + (1 - e^{-\Gamma t})\mathsf{p}(t)\big], \tag{27}$$

$$\bar{\mathsf{p}}(t) = -\mathsf{p}(t). \tag{28}$$

The fermionic duality (23) expresses an exact restriction on the possible parameter dependence of $\Pi(t)$ based only on the quite generic *physical assumptions* (I)–(III) mentioned at the beginning of Sec. 3 and two *fundamental physical principles*, the Pauli exclusion (anticommutation relations) and fermion-parity superselection applied to the *total system*. One may think of $\bar{\Pi}(t)$ as a continuation of $\Pi(t)$—considered as function of microscopic parameters—from a physical domain to a larger domain of unphysical values. This is not uncommon in physics, cf. for example, the complexification of angular momentum in scattering theory (Regge theory). In the present case, the system-reservoir coupling Hamiltonian is mapped to *anti-Hermitian* values, $H_T \rightarrow iH_T$. This corresponds[4] to a Wick-rotation $t_{rl\omega} \rightarrow it_{rl\omega}$ *together with* inversion of the relative sign between the two tunneling terms in $H_T$. The fermionic duality is the imprint left behind in the *reduced* description (*after* tracing out reservoirs) of the mentioned physical assumptions and principles (*before* tracing). It takes the form of a restriction (23) on the continuation *beyond* the physical parameter domain. It is not required—or to be expected—that the superoperator $\bar{\Pi}(t)$ resulting from the parameter substitution (25) should be a physical evolution. The construction as a continuation guarantees that $\bar{\Pi}(t)$ is still a TP map, but we will see that it is not CP. Nevertheless, approximations that break fermionic duality are inconsistent with the physical assumptions and principles governing the underlying *total system*, see Sec. 6. After presenting all our results we will compare with other works in the discussion [Sec. 6].

### 3.1.1 Cross-relation left and right eigenvectors

We now first explain the usefulness of relation (23), extending the analysis of Ref. [10]. It implies that if $\big|\pi_i(t)\big)$ is a *right* eigenvector of $\Pi(t)$ with eigenvalue $\pi_i(t)$ then

$$\pi_j(t) = e^{-\Gamma t}\,\bar{\pi}_i(t)^* \tag{29a}$$

is also an—in general different—eigenvalue, numbered $j \neq i$, with *left* eigenvector

$$\big(\pi_j'(t)\big| = \big(\bar{\pi}_i(t)\big|\mathcal{P} = \big[\mathcal{P}\big|\bar{\pi}_i(t)\big)\big]^\dagger. \tag{29b}$$

Similarly, right eigenvectors are related to left ones by

$$\big|\pi_j(t)\big) = \mathcal{P}\big|\bar{\pi}_i'(t)\big) = \big[\big(\bar{\pi}_i'(t)\big|\mathcal{P}\big]^\dagger. \tag{29c}$$

---

[3]Relation (27), written as $(1 - e^{-\Gamma t})\mathsf{p}(t) = \mathsf{g}(t) + e^{-\Gamma t}\bar{\mathsf{g}}(t)$, is obtained by inserting (7) on the left and partially integrating using $\mathsf{p}(0) = 0$. Relation (28) follows by taking the overbar of Eq. (27).

[4]The substitution $H_T \rightarrow iH_T$ means that we treat the conjugate pair of tunnel constants in $H_T = \sum_{rl} \int d\omega \big(t_{rl} d_l^\dagger c_{r\omega} + t_{rl}^* c_{r\omega}^\dagger d_l\big)$ as independent parameters: $t_{rl} \rightarrow it_{rl}$ but $t_{rl}^* \rightarrow it_{rl}^* = (-it_{rl})^*$. This inverts the sign of all spectral densities $t_{rl}t_{r'l'}^* \rightarrow -t_{rl}t_{r'l'}^*$ determining the decay rates, see Ref. [10].

Thus, although $\Pi(t)$ is *not* a unitary matrix [cf. Eq. (1)] its left and right eigenvectors are nevertheless related by conjugation up to parity signs ($\mathcal{P}$) and a parameter substitution (25) (overbar).

The duality only ensures proportionality of the vectors in Eqs. (29b)–(29c). The proportionality constants were chosen such that binormalization imposed for pair $i$ is preserved for pair $j$: $(\pi'_j(t)|\pi_j(t)) = (\bar{\pi}'_i(t)|\bar{\pi}_i(t))^* = 1$. One is then still free to gauge the right hand side of Eq. (29c) by any nonzero time-dependent complex scalar $\theta_j(t)$ and correspondingly Eq. (29b) by $1/\theta_j(t)$. If an eigenvalue happens to be *self-dual*, $\pi_i(t) = e^{-\Gamma t} \bar{\pi}_i(t)^*$, we have $i = j$ in Eq. (29a). In this case the gauge freedom is fixed by binormalization $(\pi'_i(t)|\pi_i(t)) = 1$:

$$\left(\pi'_i(t)\right| = \frac{1}{\left(\bar{\pi}_i(t)\big|\mathcal{P}\big|\pi_i(t)\right)} \left(\bar{\pi}_i(t)\big|\mathcal{P}, \qquad \big|\pi_i(t)\right) = \frac{1}{\left(\pi'_i(t)\big|\mathcal{P}\big|\bar{\pi}'_i(t)\right)} \mathcal{P}\big|\bar{\pi}'_i(t)\right), \qquad (30)$$

with related factors $\left(\bar{\pi}_i(t)\big|\mathcal{P}\big|\pi_i(t)\right) \cdot \left(\pi'_i(t)\big|\mathcal{P}\big|\bar{\pi}'_i(t)\right) = 1$.

Table 2 shows that for the resonant level model all eigenvalues are indeed *cross*-related by the duality relation (29a). The nontrivial, non-exponential time-dependence is located in the eigen*vectors*. The duality relation (29b) now dictates that if the right eigenvector to eigenvalue $\pi_0(t) = 1$ depends nontrivially on time through $\mathsf{p}(t)$, then the same *must* hold for the left eigenvector to eigenvalue $\pi_3(t) = e^{-\Gamma t}$, see Table 2 and Eq. (28). Analogously, the time-constancy of the left and right eigenvectors $i = 1$ dictates the time-constancy of the $i = 2$ eigenvectors. Thus, duality provides a fine-grained insight into the location of nontrivial contributions to the dynamics.

In an analytical calculation of the spectrum of $\Pi(t)$ one may, for example, determine for each dual pair only one eigenvalue and its left and right eigenvector algebraically, and then obtain the remaining eigenvalues and eigenvectors via a mere parameter substitution [Eq. (25)] and parity transform $\mathcal{P}$ [Eq. (24)]. This is much simpler and, moreover, preserves the compactness of analytical expressions already obtained. For models only slightly more complicated than the resonant level this already leads to significant simplifications and some surprising insights as shown in the weak coupling limit [10, 15–17].

### 3.1.2 Constraints on evolution of states and observables

We have seen for the resonant level that the duality (29) dictates that terms with qualitatively similar time-dependence in the spectral decomposition of $\Pi(t)$ occur pairwise on opposite ends of the real part of the eigenspectrum. In the general dynamics,

$$\big|\rho(t)\big) = \Pi(t)\big|\rho(0)\big) = \sum_{i=0}^{n} \pi_i(t)\big|\pi_i(t)\big)\left(\pi'_i(t)|\rho(0)\right), \qquad (31)$$

one pair of contributions is of particular interest.

The right eigenvector to eigenvalue $\pi_0 = 1$ is a *time-dependent fixed point*[5], $\Pi(t)\big|\pi_0(t)\big) = \big|\pi_0(t)\big)$, which is guaranteed to exist by the evolution's TP property, $\left(\pi'_0\big|\Pi(t) = \left(\pi'_0\right|\right.$ writing $\left(\pi'_0\right| = \mathrm{Tr}$. Often the operator $\big|\pi_0(t)\big)$ is unique and can then be scaled to a positive, trace-normalized *physical state*[6]. For simplicity we assume throughout the paper that the eigenvalue $\pi_0$ is nondegenerate. The time-dependence of the fixed-point is important and its significance was recently highlighted [23]: If one initially prepares $\big|\rho(0)\big) = \big|\pi_0(t_\mathrm{r})\big)$ where the reentrance time $t_\mathrm{r} > 0$ is a parameter, then the nontrivial evolution is guaranteed to *exactly* recover this state at the preset time $t = t_\mathrm{r}$, $\Pi(t_\mathrm{r})\big|\rho(0)\big) = \big|\pi_0(t_\mathrm{r})\big)$, even though the *environment* state for

---

[5]For a given time, the fixed point of a dynamical map relates to the disturbance caused by its measurement operators [63].

[6]See Chap. 6. of Ref. [64] and discussion in Ref. [23].

$t = 0$ generally differs from the one at $t = t_r$. For the resonant level model this reentrant behavior signals the breakdown of semigroup-Markovianity [23].

The duality cross-relation (29a) now dictates that the dynamics has another fundamental eigenvalue $\pi_n(t) = e^{-\Gamma t}$ with trivial time-dependence at the opposite end of the spectrum. Here we number $i = 0, \ldots, n$ where $n := d^2 - 1$ and $d$ is the system Hilbert space dimension. The right eigenvector $\left|\pi_n\right) = (-\mathbb{1})^N$ is the time-constant parity operator,

$$\Pi(t)\left|(-\mathbb{1})^N\right) = e^{-\Gamma t}\left|(-\mathbb{1})^N\right). \tag{32}$$

We note that this follows directly from the fact that the dual propagator $\bar{\Pi}(t)$ is also a TP map[7], $\left(\mathbb{1}\middle|\bar{\Pi}(t) = \left(\mathbb{1}\middle|\right.\right.$. The corresponding left eigenvector can be expressed via the zeroth right eigenvector, $\left(\pi_n'(t)\middle| = \text{Tr}\{\bar{\pi}_0(t)(-\mathbb{1})^N \bullet\}\right.$, where $\bar{\pi}_0(t)$ denotes the self-adjoint operator specifying $\left|\bar{\pi}_0(t)\right)$. It determines the amplitude in the expansion of the time-dependent state:

$$\left|\rho(t)\right) = \left|\pi_0(t)\right) + \ldots + e^{-\Gamma t}\,\text{Tr}\{\bar{\pi}_0(t)(-\mathbb{1})^N \rho(0)\} \cdot \left|(-\mathbb{1})^N\right). \tag{33}$$

Thus, the nontrivial time-dependence of the coefficient of the fast $\Gamma$-decay is *also* determined by the time-dependent *non-decaying fixed-point* component $\left|\pi_0(t)\right)$, namely through its functional dependence on parameters. For semigroup dynamics this coefficient is time-constant, but in general it is time-dependent, even in the resonant level model, see $\left|\pi_0(t)\right)$ in Table 2. The result (33) implies that the expectation value of a system observable $A$ can be decomposed into an instantaneous expectation value in the time-dependent *fixed-point state* plus corrections:

$$\langle A\rangle_{\rho(t)} = \langle A\rangle_{\pi_0(t)} + \ldots + e^{-\Gamma t}\,\text{Tr}\{\bar{\pi}_0(t)(-\mathbb{1})^N \rho(0)\} \cdot \text{Tr}\{A(-\mathbb{1})^N\}. \tag{34}$$

The corrections with the fast $\Gamma$-decay appear only for observables which overlap with the fermion-parity, $\text{Tr}\{A(-\mathbb{1})^N\} \neq 0$. Such operators $A$ depend multiplicatively on the occupations of all fermionic orbitals in the open system, i.e., they probe global correlations within the system. The above insight into the general dynamics extends the weak-coupling results of Ref. [16].

## 3.2 Measurement operator sum

We now turn to an entirely different formulation of the same dynamics which is ubiquitous in quantum information theory. We can apply this approach here since we are assured that $\Pi(t)$—being the exact evolution—is a CP map[8]. It can therefore be written in the form of a Sudarshan-Kraus operator sum [20–22]

$$\Pi(t) = \sum_\alpha m_\alpha(t) M_\alpha(t) \bullet M_\alpha(t)^\dagger. \tag{35}$$

Without loss of generality we choose to normalize the *measurement operators* using the Hilbert-Schmidt scalar product, $\text{Tr}\,M_\alpha(t)^\dagger M_\alpha(t) = 1$. The coefficients $m_\alpha(t)$ are then real and positive by CP, see App. A, and the TP property of $\Pi(t)$ is equivalent to

$$\sum_\alpha m_\alpha(t) M_\alpha(t)^\dagger M_\alpha(t) = \mathbb{1}. \tag{36}$$

By taking the trace this implies a scalar sum rule: the coefficients must sum to the Hilbert space dimension $d$,

$$\sum_\alpha m_\alpha(t) = d. \tag{37}$$

---

[7]This follows from $\left[\Pi(t)\left|(-\mathbb{1})^N\right)\right]^\dagger = \left(\mathbb{1}\middle|\mathcal{P}\Pi(t)^\dagger \stackrel{(23)}{=} e^{-\Gamma t}\left(\mathbb{1}\middle|\bar{\Pi}(t)\mathcal{P} = e^{-\Gamma t}\left((-\mathbb{1})^N\middle|\right.\right.\right.$.

[8]The CP property is very difficult to maintain when performing approximations, see Ref. [33] for a discussion and references.

Each term in the operator sum (35) describes a physical process in which outcome $\alpha$ is obtained by a measurement on the environment R in some basis. For each different choice of a basis, there is a set of measurement operators $\{M_\alpha(t)\}$ and thus a different operator-sum representation. We fix this freedom by considering canonical measurement operators which are orthonormal, $\text{Tr}\, M_\alpha(t)^\dagger M_{\alpha'}(t) = \delta_{\alpha\alpha'}$. If the $m_\alpha(t)$ are nondegenerate, this fixes the set $\{M_\alpha(t)\}$ uniquely up to trivial changes by phase factors which cancel out term-by-term in the sum (35), see App. A for the case of degeneracy. Importantly, for *fermionic* systems the operators must have a definite parity denoted by $(-1)^{N_\alpha}$, i.e., $(-\mathbb{1})^N M_\alpha (-\mathbb{1})^N = (-1)^{N_\alpha} M_\alpha$, since the operators describe measurements[9].

### 3.2.1 Cross-relation of Heisenberg and Schrödinger measurement operators

From the operator sum (35) it is easy to find the measurement operators for the Heisenberg evolution $\Pi^H(t) = \Pi(t)^\dagger$ by using Eq. (22),

$$\Pi(t)^\dagger = \sum_\alpha m_\alpha(t) M_\alpha(t)^\dagger \bullet M_\alpha(t). \tag{38}$$

To see the nontrivial implication of fermionic duality (23), we insert Eq. (35) and Eq. (38) and show that the individual terms in the two operator sums must be equal up to a permutation of the summation index $\alpha$. This follows most elegantly by the Choi-Jamiołkowski (CJ) correspondence for which the fermionic duality is worked out in App. A. We obtain the key result that pairs of orthonormal measurement operators with the same parity obey

$$M_\alpha(t)^\dagger = \bar{M}_{\alpha'}(t), \tag{39a}$$

and their corresponding coefficients fulfill

$$m_\alpha(t) = e^{-\Gamma t}(-1)^{N_{\alpha'}} \bar{m}_{\alpha'}(t), \tag{39b}$$

where $\alpha = \alpha'$ is allowed. In Eq. (39a) the only freedom left in the relation between the operators $M_\alpha$ and $M_{\alpha'}$ is a complex phase factor, which we set to 1.

The fermionic duality relation (39) implies that if a coefficient is self-dual, $m_\alpha = e^{-\Gamma t}(-1)^{N_\alpha}\bar{m}_\alpha$, the measurement operator is a strongly constrained function: its adjoint must correspond to dual parameters, $M_\alpha^\dagger = \bar{M}_\alpha$. In all other cases, for each pair $\alpha$, $\alpha'$ of dual coefficients one needs to determine only one of the measurement operators, obtaining its dual operator *for free*. Thus, very similar to the relation (29) between left and right eigenvectors of $\Pi(t)$, the difficult task of analytically finding the measurement operators and coefficients for nontrivial fermionic open systems is significantly simplified.

### 3.2.2 Additional fermionic sum rule for measurement operators

Since the dual propagator $\bar{\Pi}(t) = \sum_\alpha \bar{m}_\alpha \bar{M}_\alpha(t) \bullet \bar{M}_\alpha(t)^\dagger$ is also a TP map, the dual measurement operators also obey a sum rule: $\sum_\alpha \bar{m}_\alpha \bar{M}_\alpha(t)^\dagger \bar{M}_\alpha(t) = \mathbb{1}$. Notably, this is *not* an obvious consequence of the TP sum rule (36) for $\Pi(t)$: inserting Eq. (39) we instead find[10] that the

---

[9]If the parity is initially definite, $[\rho, (-\mathbb{1})^N] = 0$, then for individual processes conditioned on outcome $\alpha$, parity is still well-defined, $[M_\alpha \rho M_\alpha^\dagger, (-\mathbb{1})^N] = 0$. This holds for any $\rho$, giving $M_\alpha (-\mathbb{1})^N \propto (-\mathbb{1})^N M_\alpha$. Applying this twice we find that the proportionality constant is some sign $(-1)^{N_\alpha}$. See also App. A.

[10]To this end, multiply $\Pi(t)(-\mathbb{1})^N = \sum_\alpha m_\alpha M_\alpha(-\mathbb{1})^N M_\alpha^\dagger = e^{-\Gamma t}(-\mathbb{1})^N$ by $(-\mathbb{1})^N$ and use $(-\mathbb{1})^N M_\alpha (-\mathbb{1})^N = (-1)^{N_\alpha} M_\alpha$. Note that not every eigenvalue equation $\Pi(t)X = \sum_\alpha m_\alpha M_\alpha X M_\alpha^\dagger = \lambda X$ can be converted to a sum rule of this form: it requires that the operator $X$ is invertible and commutes up to a scalar factor with all measurement operators $M_\alpha X = k_\alpha X M_\alpha$.

original measurement operators of the fermionic systems must obey an *additional, independent* sum rule:

$$\sum_\alpha (-1)^{N_\alpha} m_\alpha(t) M_\alpha(t) M_\alpha(t)^\dagger = e^{-\Gamma t} \mathbb{1}. \tag{40}$$

This shares with Eq. (32) the remarkable feature of depending only on a *single* detail of the microscopic model, the lump sum of couplings $\Gamma$, independent of interactions and external controls such as temperature, and chemical potentials. Unlike the familiar sum rule (36), the adjoint appears on the *right* operator and the *difference* of even and odd parity terms is constrained to a *time-dependent* operator.

The trace of Eq. (40) implies an extra scalar sum rule

$$\sum_\alpha (-1)^{N_\alpha} m_\alpha(t) = d\, e^{-\Gamma t}, \tag{41}$$

where we used that $\mathrm{Tr}\, M_\alpha(t)^\dagger M_{\alpha'}(t) = \delta_{\alpha\alpha'}$ for canonical measurement operators. Together with Eq. (37) we obtain separate sum rules for the coefficients of the even and odd measurement-operators as functions of time:

$$\sum_\alpha \tfrac{1}{2}[1 + (-1)^{N_\alpha}] m_\alpha(t) = d\, \tfrac{1}{2}[1 + e^{-\Gamma t}], \tag{42a}$$

$$\sum_\alpha \tfrac{1}{2}[1 - (-1)^{N_\alpha}] m_\alpha(t) = d\, \tfrac{1}{2}[1 - e^{-\Gamma t}]. \tag{42b}$$

Whereas for $t = 0$ the even operators must contain all the weight to produce $\Pi(0) = \mathbb{1} \bullet \mathbb{1}$, the even and odd weights coincide in the stationary limit $t \to \infty$, evenly splitting the standard sum rule (37). In other words, the stationary evolution gives equal weight to parity changing and parity preserving processes.

For the resonant level model the measurement operators are indexed by $\alpha = N\eta$ with level occupation $N = 0, 1$ for even or odd parity and $\eta = \pm$. The operators (12) and coefficients (13) have a simple explicit dependence on $\Gamma, \varepsilon$. All nontrivial parameter dependence is contained in the function $\mathsf{p}(t)$ [Eq. (7)] which has a simple transform $\bar{\mathsf{p}}(t) = -\mathsf{p}(t)$ [Eq. (28)] under the parameter mapping $(\varepsilon, \mu, \Gamma) \to (-\varepsilon, -\mu, -\Gamma)$. Thus, duality strongly restricts the functional form of the coefficients (13) for even parity, and for odd parity pairs them up:

$$m_{0\eta} = +e^{-\Gamma t} \bar{m}_{0\eta}, \qquad\qquad m_{1\eta} = -e^{-\Gamma t} \bar{m}_{1(-\eta)}. \tag{43}$$

Correspondingly, the even-parity operators (12) are *self-dual* and the odd ones are dual partners,

$$M_{0\eta}^\dagger = \bar{M}_{0\eta}, \qquad\qquad M_{1\eta}^\dagger = \bar{M}_{1(-\eta)}. \tag{44}$$

The *self-duality* strongly constraints the coefficients of $d^\dagger d$ and $d d^\dagger$ inside the operator $M_{0\eta}^\dagger$: the substitution $(\varepsilon, \mu, \Gamma) \to (-\varepsilon, -\mu, -\Gamma)$ maps the coefficients to their complex conjugates. We stress that *without* the unphysical inversion of the decay rates $\Gamma \to -\Gamma$ one cannot explain this puzzling "symmetry" of this exact result. Furthermore, it is by no means obvious from the explicit solutions (12)–(13) that the additional simple sum rule (40) indeed holds. Also, the scalar sum rule (42) implies for fixed level occupation $N = 0$ or 1 that any nontrivial time-dependence of the coefficients (13) for $\eta = \pm$ must be the same up to a sign. All these structural features of the measurement operators were left unexplained in Ref. [23].

### 3.2.3 Unphysicality of the duality mapping

Of all the approaches to be discussed, the measurement-operator formulation (39) most clearly reveals that the dual propagator $\bar{\Pi}(t)$ is *unphysical*. It is *not CP* whenever $\Pi(t)$ and $\Pi^{\mathrm{H}}(t)$ are CP: the fermion-parity signs in Eq. (39b) imply that for operators $\bar{M}_\alpha$ with odd-parity the coefficients $\bar{m}_{\alpha'}$ are strictly negative. This means that due to the inversion of coupling constants, $\Gamma \mapsto -\Gamma$, $\bar{\Pi}(t)$ cannot correspond to the evolution of *any* physical system. In the duality relation (23) this is reflected in the parity transformation $\mathcal{P}$.

We stress that this unphysicality in no way obstructs the derivation of Eq. (23) or its useful application to physical problems. On the contrary, it makes the duality mapping particularly interesting: by continuation of parameters to non-physical domains [Eq. (28) ff.] it points out functional dependencies which are not just physically "unintuitive" but even *impossible* to motivate by strictly physical parameter mappings.

## 4 Fermionic duality for exact quantum master equations

We now consider how fermionic duality constrains equivalent exact quantum master equations which *generate* the evolution $\Pi(t)$.

### 4.1 Time-local quantum master equation

The dynamics can be described by a time-local QME [5]

$$\frac{d}{dt}\Pi(t) = -i\mathcal{G}(t)\Pi(t), \qquad\qquad \Pi(0) = \mathcal{I}. \qquad (45)$$

Importantly, the time-local generator $\mathcal{G}(t)$ is in general *time-dependent* even though it derives microscopically from a *time-constant* Hamiltonian generator $H_{\mathrm{tot}}$ for system plus reservoirs. The generator of the corresponding Heisenberg evolution acts from the left,

$$\frac{d}{dt}\Pi(t)^\dagger := i\mathcal{G}^{\mathrm{H}}(t)\Pi(t)^\dagger, \qquad (46)$$

in order to generate the evolution of observables (21) as $\frac{d}{dt}A(t) = i\mathcal{G}^{\mathrm{H}}(t)A(t)$. As a consequence, it is *not*[11] simply equal to the adjoint of the generator:

$$\mathcal{G}^{\mathrm{H}}(t) = \left[\Pi(t)^{-1}\mathcal{G}(t)\Pi(t)\right]^\dagger \neq \mathcal{G}(t)^\dagger. \qquad (47)$$

This difference implies that for open systems one cannot switch from the equation of motion in the Schrödinger picture, $\frac{d}{dt}\rho(t) = -i\mathcal{G}(t)\rho(t)$, to the Heisenberg picture, $\frac{d}{dt}A(t) = i\mathcal{G}^{\mathrm{H}}(t)A(t)$, *without* first *solving* the dynamics. This is a known complication (Ref. [4], p. 125) of the analysis of open-system evolutions not commuting with their generator [65].

This nontrivial problem—specific to open systems—is *solved by fermionic duality* and will play a role in the construction of approximations in Sec. 5. Only in the simple cases where $[\mathcal{G}(t), \Pi(t)] = 0$ do we have $\mathcal{G}^{\mathrm{H}}(t) = \mathcal{G}(t)^\dagger$. This includes the familiar case of Markovian semigroup dynamics where a time-constant $\mathcal{G}$ generates $\Pi(t) = e^{-i\mathcal{G}t}$. However, already for the resonant level model we have $\mathcal{G}^{\mathrm{H}}(t) \neq \mathcal{G}(t)^\dagger$ since time-ordering of the generator matters except for special parameters ($T = \infty$ or $\varepsilon = \mu$, see Sec. 2).

---

[11]The adjoint equation $\frac{d}{dt}\Pi(t)^\dagger := i\Pi(t)^\dagger\mathcal{G}(t)^\dagger$ suggests to identify $\mathcal{G}(t)^\dagger$ with the generator. However, since it acts on the *right* one verifies that it is not the generator in the equation of motion for an observable $A(t) := \Pi(t)^\dagger A(0)$ which is instead $\mathcal{G}^{\mathrm{H}}(t)$.

The TP property of the Schrödinger evolution, $\mathrm{Tr}\,\mathcal{G}(t) = 0$, by Eq. (47), corresponds to the Heisenberg evolution being unit-preserving or unital,

$$\mathcal{G}^{\mathrm{H}}(t)\big|\mathbb{1}\big) = \mathcal{G}^{\mathrm{H}}(t)\Pi(t)^{\dagger}\big|\mathbb{1}\big) = \big[\big(\mathbb{1}\big|\mathcal{G}(t)\Pi(t)\big]^{\dagger} = 0 \,. \tag{48}$$

Physically this means that trivial measurements stay trivial.

Taking the time-derivative of relation (23) one obtains the fermionic duality for the *time-local* generator:

$$\mathcal{G}^{\mathrm{H}}(t) = \big[\Pi(t)^{-1}\mathcal{G}(t)\Pi(t)\big]^{\dagger} = i\Gamma\,\mathcal{I} - \mathcal{P}\bar{\mathcal{G}}(t)\mathcal{P}\,. \tag{49}$$

This relation is another key result of the paper which we again stress is exact, in particular, it is *not* based on any time-local approximation. It solves the nontrivial task of obtaining the Heisenberg generator $\mathcal{G}^{\mathrm{H}}(t)$ directly from $\mathcal{G}(t)$, without computing $\Pi(t)$. Already for the resonant level model this presents a significant simplification: instead of performing quite some superoperator algebra[12] as required by Eq. (47), we obtain $\mathcal{G}^{\mathrm{H}}(t)$ by the simple parameter substitution $\mathrm{g}(t) \to \bar{\mathrm{g}}(t)$ given in Eq. (27).

### 4.1.1 Cross-relation left and right eigenvectors of the generator

The time-local fermionic duality (49) immediately implies that if $\big|g_i(t)\big)$ is a *right* eigenvector of $\mathcal{G}(t)$ with eigenvalue $g_i(t)$ then

$$g_j(t) = \big[i\Gamma - \bar{g}_i(t)\big]^* \tag{50a}$$

is also a—generally different—eigenvalue $g_j$ with *left* eigenvector

$$e^{\Gamma t/2}\big(g_j'(t)\big| = \big(\bar{g}_i(t)\big|\mathcal{P}\Pi(t)^{-1} = \Big[(\Pi(t)^{-1})^{\dagger}\mathcal{P}\big|\bar{g}_i(t)\big)\Big]^{\dagger}\,. \tag{50b}$$

Similarly, for right eigenvectors:

$$e^{-\Gamma t/2}\big|g_j(t)\big) = \Pi(t)\mathcal{P}\big|\bar{g}_i'(t)\big) = \Big[\big(\bar{g}_i'(t)\big|\mathcal{P}\Pi(t)^{\dagger}\Big]^{\dagger}\,. \tag{50c}$$

As before [Eq. (29) ff.], the proportionality factors were chosen to ensure that the binormalization of pair $i$ is passed on to pair $j$: $\big(g_j'(t)|g_j(t)\big) = \big(\bar{g}_i'(t)|\bar{g}_i(t)\big)^* = 1$. For self-dual eigenvalues $g_i(t) = [i\Gamma - \bar{g}_i(t)]^*$ biorthonormality $\big(g_i'(t)|g_i(t)\big) = 1$ implies

$$\big(g_i'(t)\big| = \frac{1}{\big(\bar{g}_i(t)\big|\mathcal{P}\Pi(t)^{-1}\big|g_i(t)\big)}\big(\bar{g}_i(t)\big|\mathcal{P}\Pi(t)^{-1}\,, \tag{51a}$$

$$\big|g_i(t)\big) = \frac{1}{\big(g_i'(t)\big|\Pi(t)\mathcal{P}\big|\bar{g}_i'(t)\big)}\Pi(t)\mathcal{P}\big|\bar{g}_i'(t)\big)\,, \tag{51b}$$

where $\big(\bar{g}_i(t)\big|\mathcal{P}\Pi(t)^{-1}\big|g_i(t)\big) \cdot \big(g_i'(t)\big|\Pi(t)\mathcal{P}\big|\bar{g}_i'(t)\big) = 1$. It is expected that fermionic duality takes a more complicated form here since the generator $\mathcal{G}(t)$ incorporates a great deal of the complexity of the solution $\Pi(t)$ into the QME (45) in order to eliminate the memory integral of QME (15). In this respect, the simplicity of the eigenvalue duality (50a) is surprising and presents a definite advantage for analytical calculations. It generalizes the relations of Refs. [10, 16] which for weak coupling imply that $\Gamma$ is always the largest decay rate [16].

As expected the cross-relation (50b)–(50c) of the eigenvectors is more complicated due to the involvement of $\Pi(t)$. Only the special case $[\mathcal{G}(t), \Pi(t)] = 0$ leads to a simpler duality relation $\mathcal{G}(t)^{\dagger} = i\Gamma\,\mathcal{I} - \mathcal{P}\bar{\mathcal{G}}(t)\mathcal{P}$ that directly relates left and right eigenvectors of $\mathcal{G}(t)$:

---

[12]Insert Eq. (8) and use $\mathcal{P}\mathcal{D}_{\eta}^{\dagger}\mathcal{P} = -\mathcal{D}_{-\eta} - \mathcal{I}$, $[L, \mathcal{D}_{\eta}] = 0$, $[L, \Pi] = 0$ and the general relations $L^{\dagger} = L$, and $\mathcal{P}L\mathcal{P} = L$.

$\big(g'_j(t)\big| = \big(\bar{g}_i(t)\big|\mathcal{P}$ and $\big|g_j(t)\big) = \mathcal{P}\big|\bar{g}'_i(t)\big)$. This includes the Markovian-semigroup limit with time-constant $\mathcal{G}$, recovering the weak-coupling results of Ref. [16].

Table 2 shows that for the resonant level model, the eigenvalues of $\mathcal{G}(t)$ indeed satisfy the cross-relation (50a). Yet, since $[\mathcal{G}(t), \Pi(t)] \neq 0$ for this model, the eigenvector relations (50b)–(50c) remain nontrivial: their verification requires the transformation (27) of the function $\bar{g}(t)$ and some algebra to verify Eq. (49). We note that $\mathcal{G}(t)$ and its eigenvectors also satisfy another, simpler relation which is, however, specific to the model and not related to general principles, see App. B.

### 4.1.2 Constraints on time derivatives of states and observables

Analogous to the fermionic duality (31) for the propagator, its time-local version (49) provides general insight into where nontrivial (non-exponential) contributions occur in the dynamics. In this case it concerns the time-derivative of the state:

$$\tfrac{d}{dt}\big|\rho(t)\big) = -i\mathcal{G}(t)\Pi(t)\big|\rho(0)\big) \tag{52a}$$

$$= -i\sum_i g_i(t)\big|g_i(t)\big)\big(g'_i(t)\big|\Pi(t)\big|\rho(0)\big) \tag{52b}$$

$$= -ie^{-\Gamma t/2}{\sum_{ij}}' g_i(t)\big|g_i(t)\big)\big(\bar{g}_j(t)\big|\mathcal{P}\big|\rho(0)\big). \tag{52c}$$

The prime indicates that we sum over pairs of dual eigenvalues $i$ and $j$ keeping only one term for self-dual ones. Here there is a catch because the evaluation of Eq. (52c) requires that the normalization of $\big|g_i(t)\big)$ is known, which we implicitly fixed in the duality relation Eq. (50). This is not an issue for two important contributions which we now discuss.

The first one is the *missing* contribution: the time-dependent *zero-mode* of the generator, $\mathcal{G}(t)\big|g_0(t)\big) = 0$. Such a right eigenvector with eigenvalue $g_0(t) = 0$ always exists since by trace preservation $\big(g'_0\big|\mathcal{G}(t) = 0$, writing $\big(g'_0\big| = \mathrm{Tr} = \big(\pi'_0\big|$. At finite times $\big|g_0(t)\big)$ is distinct from the time-dependent fixed point $\big|\pi_0(t)\big)$ of $\Pi(t)$ [Eq. (31)], even though asymptotically both converge to the stationary state $\big|\rho(\infty)\big) = \big|g_0(\infty)\big) = \big|\pi_0(\infty)\big)$ whenever it is unique[13]. In fact, for the resonant level $\big|g_0(t)\big)$ even fails to be a positive operator in time intervals where $|g(t)| > 1$ [Table 2], which happens precisely in parameter regimes where the dynamics is not CP-divisible shown Fig. 1. In contrast, $\big|\pi_0(t)\big)$ is always positive since $|p(t)| \leq 1$ for all $t$, see Sec. 2.

The fermionic duality (50a) implies that there is another fundamental contribution to Eq. (52c) with eigenvalue $g_n(t) = -i\Gamma$ at the other end of the spectrum. Remarkably, it only depends on $\Gamma$ and its right eigenvector does not depend on *any* microscopic detail:

$$\mathcal{G}(t)\big|(-\mathbb{1})^N\big) = -i\Gamma\big|(-\mathbb{1})^N\big). \tag{53}$$

This follows from $\big|g_n(t)\big) = e^{\Gamma t/2}\Pi(t)\big|(-\mathbb{1})^N\big) = e^{-\Gamma t/2}\big|(-\mathbb{1})^N\big)$ [Eq. (32)]. Note that this also follows from the fact that $\bar{\mathcal{G}}(t)$ generates a TP map, $\big(\mathbb{1}\big|\bar{\mathcal{G}}(t) = 0$. The corresponding left eigenvector is $\big(g'_n(t)\big| = e^{-\Gamma t/2}\mathrm{Tr}\{\bar{g}_0(t)(-\mathbb{1})^N\Pi(t)^{-1}\bullet\}$ [Eq. (29b)] where $\bar{g}_0(t)$ denotes the self-adjoint operator specifying $\big|\bar{g}_0(t)\big)$. It inherits the parameter dependence of the zero-mode which in general is complicated,

$$\tfrac{d}{dt}\big|\rho(t)\big) = -e^{-\Gamma t}\Gamma\big|(-\mathbb{1})^N\big)\big(\bar{g}_0(t)\big|\mathcal{P}\big|\rho(0)\big) + \dots. \tag{54}$$

For the time-derivative of the expectation value of an observable $A$ the *nontrivial time-dependent* prefactor of the $\Gamma$-decay rate,

$$\tfrac{d}{dt}\langle A \rangle = -e^{-\Gamma t}\Gamma\,\mathrm{Tr}\{A(-\mathbb{1})^N\} \cdot \mathrm{Tr}\{\bar{g}_0(t)(-\mathbb{1})^N\rho(0)\} + \dots \tag{55}$$

---

[13]This agrees with the stationary state obtained from the memory kernel, $\big|\hat{k}_0(0)\big) = \big|g_0(\infty)\big)$ [Eq. (79)].

is completely determined by the parameter dependence of the *zero-mode of the generator*, $\big|\bar{g}_0\big)$, the *missing* term in the expansion (52c).

This remarkable structure is a generalization of the weak-coupling result of Ref. [16] which introduces a new distinction: whereas the fixed-point of $\Pi(t)$ determines this fast contribution to $\rho(t)$ and expectation values $\langle A \rangle(t)$ [Eq. (34)], it is the *distinct* zero-mode of $\mathcal{G}(t)$ that determines the fast contribution to $\frac{d}{dt}\rho(t)$ or *currents* $\frac{d}{dt}\langle A \rangle(t)$ [Eq. (55)]. Whereas for $[\mathcal{G}(t), \Pi(t)] = 0$ (including Markovian semigroups) the fixed point and zero mode coincide at any time, they are in general different, $\big|\bar{\pi}_0(t)\big) \neq \big|\bar{g}_0(t)\big)$. This illustrates that fermionic duality leads to *independent* insights when formulated in complementary approaches.

For the simple resonant level model this leads to an interesting insight into the *transport current* by taking $A = N$, the level occupation operator. Using Eq. (52c) one can verify that the omitted terms in Eq. (55) are zero because $(N|g_j(t)) = 0$ except for $j = 3$. The normalization of the eigenvector $\big|g_3(t)\big)$ can then be calculated[14] by taking the trace of Eq. (50c) and we obtain from Eq. (55)

$$\frac{d}{dt}\langle N \rangle = \Gamma e^{-\Gamma t} \tfrac{1}{2}\left[\bar{g}(t) + ((-1)^N|\rho(0))\right]. \tag{56}$$

Thus, the observable transport current is *automatically* decomposed in two contributions of which one is a trivial exponential decay depending on the initial state through $\tfrac{1}{2}((-1)^N|\rho(0)) = \tfrac{1}{2} - \langle N \rangle(0)$. All nontrivial time-dependence is captured by the *single* function $g(t)$ from the generator $\mathcal{G}(t)$ but evaluated at *dual parameters*. Note that this relation does not follow from Eq. (34) unless one laboriously uses identities connecting the nontrivial functions $g$ and $p$.

## 4.2 Jump operator sum

As mentioned in the introduction, a distinct advantage of the previous time-local QME approach is that it connects to the *divisibility* properties of the dynamics (Markovianity). These are, however, only revealed when decomposing the generators $\mathcal{G}$ and $\mathcal{G}^{\mathrm{H}}(t)$ [Eq. (47)] appearing in the time-local fermionic duality (49) into *jump-operator* sums, analogous to the decomposition of $\Pi(t)$ into a measurement-operator-sum.

### 4.2.1 Causal and anti-causal divisibility

This approach requires some preliminary discussion. We first note that the Hermicity- and trace-preservation properties of the dynamics *alone* already imply the following structure of the generator [App. C] due to Lindblad, Gorini, Kossakowski and Sudarshan [24, 25]

$$-i\mathcal{G}(t) = -i[H(t), \bullet] + \sum_{\alpha} j_{\alpha}(t)\mathcal{D}_{\alpha}(t). \tag{57}$$

The dissipators $\mathcal{D}_{\alpha}(t) = J_{\alpha}(t)\bullet J_{\alpha}(t)^{\dagger} - \tfrac{1}{2}\left\{J_{\alpha}(t)^{\dagger}J_{\alpha}(t), \bullet\right\}$ contain jump operators $J_{\alpha}(t)$ and are weighted with real coefficients $j_{\alpha}(t)$ which we assume to be nondegenerate, see App. C for the degenerate case. Their structure guarantees that the generated dynamics is TP ($\mathrm{Tr}\,\mathcal{D}_{\alpha}(t) = 0$). The coefficients $j_{\alpha}(t)$ need not be positive, in contrast to the coefficients of measurement operators [Eq. (37)]. The effective Hamiltonian $H(t)$ is Hermitian but differs from the bare one, $H$, which we indicate by the time argument.

Similar to the measurement operators [Sec. 3.2], we eliminate gauge freedom by working with canonical jump operators, which are orthonormal, both mutually $\mathrm{Tr}\,J_{\alpha}(t)^{\dagger}J_{\beta}(t) = \delta_{\alpha\beta}$ and to the identity, $\mathrm{Tr}\,J_{\alpha}(t) = 0$. Importantly, the canonical jump operators $J_{\alpha}(t)$ have a definite parity $(-1)^{N_{\alpha}}$, i.e. $(-1)^N J_{\alpha}(t)(-1)^N = (-1)^{N_{\alpha}}J_{\alpha}(t)$ and the canonical effective Hamiltonian has even parity, $(-1)^N H(t)(-1)^N = H(t)$ [App. C].

---

[14]In this case we can circumvent the calculation of $\Pi(t)$ in Eq. (50c) because we only need the normalization of $\big|g_0(t)\big)$ which can be fixed using the known left eigenvector $\big(\mathbb{1}\big|\Pi(t) = \big(\mathbb{1}\big|$.

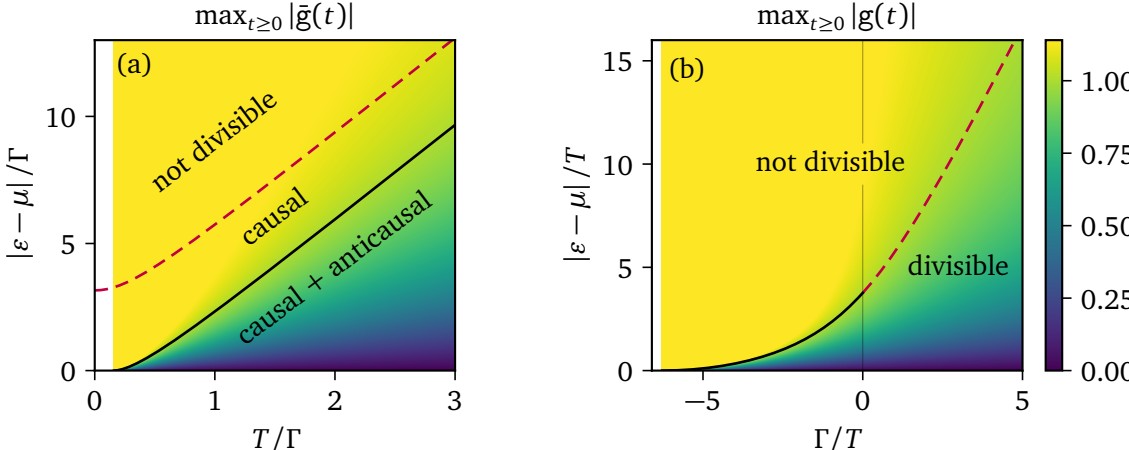

Figure 2: Regions of causal and anti-causal CP-divisibility for the resonant level model. The Heisenberg jump rates $j_\eta^{\mathrm{H}}(t) = \frac{1}{2}[1 - \eta\bar{g}(t)]\Gamma$ are positive if and only if $|\bar{g}(t)| \leq 1$. (a) $\max_{t\geq 0} |\bar{g}(t)|$ as function of level detuning $\varepsilon - \mu$ and temperature $T$. The black line marks $\max_{t\geq 0} |\bar{g}(t)| = 1$, forming the boundary of the anti-causally divisible region. The dashed line limits the causally divisible regime by marking $\max_{t\geq 0} |g(t)| = 1$, see Fig. 1. (b) Continuation of $\max_{t\geq 0} |g(t)|$ from physical ($\Gamma > 0$) to dual parameters ($\Gamma < 0$) shows the connection of causal and anti-causal divisibility revealed by fermionic duality. In (a) this connection is hidden in the $T/\Gamma \to \infty$ limit. At resonance ($\varepsilon = \mu$) the evolution is a semigroup which is trivially both causally and anti-causally divisible. As one tunes further away from resonance, the evolution first looses the anti-causal and then the causal divisibility. For $T < \Gamma/(2\pi)$ (white regions) $\max_{t\geq 0} |\bar{g}(t)| = \infty$, reflecting that the stationary limit of $\mathcal{G}^{\mathrm{H}}(t)$ does not exist even though the limit $\Pi^{\mathrm{H}}(\infty)$ is well defined. This is a peculiarity of the time-local description.

The dynamics is CP divisible, $\Pi(t) = \Pi(t, t')\Pi(t')$ for all $0 \leq t' \leq t \leq \infty$, if and only if the condition[15] $j_\alpha(t) \geq 0$ holds for all $\alpha$ and $t \geq 0$ [35, 36]. In this case the jump operators have an operational meaning: $J_\alpha(t)$ is a measurement operator for outcome $\alpha$ measured on the environment with infinitesimal probability $j_\alpha(t)\delta t \geq 0$ during infinitesimal evolution. For example, in the resonant level model, the two jump rates $j_\eta(t) = \frac{1}{2}[1 - \eta g(t)]\Gamma$ are positive if and only if $|g(t)| \leq 1$ which holds true for the parameter in the divisible region mapped out in Figs. 1 and 2(a). In this case the corresponding odd-parity jump-operators $J_\eta = d_\eta$ [cf. Eq. (11)] represent a jump of a particle to or from the level induced by a measurement in the environment in an infinitesimal time $\delta t$. The effective Hamiltonian coincides with the original one, $H(t) = H = \varepsilon d^\dagger d$ [Eq. (18a)]. The relation to stochastic simulation methods will be discussed in Sec. 6.

Divisibility also has a clear operational meaning in terms of a simulation task. The condition states that the full evolution up to time $t$ can be simulated by stopping the evolution earlier at $t'$—decoupling and discarding the environment—and then applying to the output some postprocessing device described by $\Pi(t, t')$. Such a physical device exists if and only if the latter is a CP map, see Sec. III of Ref. [23] for a discussion. If such a simulation is possible for every $t'$ and every $t$, then $\Pi(t)$ is called CP divisible. This indicates that the input-output correlations of the dynamics are weak. For this purpose we only need to inquire into the

---

[15]If $\mathcal{G}(t)$ satisfies this condition of CP-divisibility, it implies that $\Pi(t) = \Pi(t, 0)$ is CP. Note that if $\mathcal{G}(t)$ does not satisfy this condition, it is not known which sufficient conditions the $J_\alpha(t)$ and $j_\alpha(t)$ should satisfy to ensure that $\Pi(t)$ is CP.

*possibility* of such a simulation, *not* its implementation.

To derive a fermionic duality for jump coefficients and operators, we need to decompose the Heisenberg generator $\mathcal{G}^{\mathrm{H}}(t)$ appearing in Eq. (49) in a similar way,

$$i\mathcal{G}^{\mathrm{H}}(t) = i[H^{\mathrm{H}}(t), \bullet] + \sum_{\alpha} j_{\alpha}^{\mathrm{H}}(t)\mathcal{D}_{\alpha}^{\mathrm{H}}(t), \tag{58}$$

with Hermitian $H^{\mathrm{H}}(t)$. The different structure of the Heisenberg dissipator, $\mathcal{D}_{\alpha}^{\mathrm{H}}(t) :=$ $J_{\alpha}^{\mathrm{H}}(t)\bullet J_{\alpha}^{\mathrm{H}}(t)^{\dagger} - \frac{1}{2}\{J_{\alpha}^{\mathrm{H}}(t)J_{\alpha}^{\mathrm{H}}(t)^{\dagger}, \bullet\}$, now ensures that the Heisenberg evolution is unit-preserving [Eq. (48)]. Moreover, the coefficients $j_{\alpha}^{\mathrm{H}}(t)$ are distinct from the $j_{\alpha}(t)$ and related to a *different* type of divisibility: $j_{\alpha}^{\mathrm{H}}(t) \geq 0$ is the condition[16] for what can be called *anti-causal CP divisibility* of the state dynamics, $\Pi(t) = \Pi(t')\Pi_{\mathrm{a}}(t, t')$ for all $t' \in [0, t]$ by some CP-TP map $\Pi_{\mathrm{a}}(t, t')$ on the *right*, in contrast to the usual division of the dynamics by postprocessing to the *left*. Whereas semigroup dynamics is both causally and anti-causally CP divisible, this does not hold for more general dynamics as studied here. For the resonant level model the parameter regime of anti-causal divisibility is mapped out in Fig. 2(a) and does *not* coincide with the regimes of causal divisibility.

The operational meaning of anti-causal divisibility becomes clear when viewed as a simulation task: The condition states that the full evolution $\Pi(t)$ up to time $t$ can be simulated by *preprocessing* its input by some device described by $\Pi_{\mathrm{a}}(t, t')$, and then afterwards running the evolution $\Pi(t')$ only up to time $t'$. Also here, a physical preprocessing device exists if and only if $\Pi_{\mathrm{a}}(t, t')$ is a CP map. If such a simulation is possible for every $t'$ and every $t$, the evolution can be called anti-causally CP divisible. This indicates that the input-output correlations of the dynamics are weak and *additionally* that the causal ordering is weak, i.e., the dynamics is robust against interruption at $t'$ and reversal of causal ordering. As for causal divisibility, we only inquire into the possibility of such a simulation, *not* its implementation.

These two types of divisibility are not related in an obvious way. It is in general possible to express $H^{\mathrm{H}}(t)$, $J_{\alpha}^{\mathrm{H}}(t)$ and $j_{\alpha}^{\mathrm{H}}(t)$ in $H(t)$, $J_{\alpha}(t)$ and $j_{\alpha}(t)$ using the measurement operators $M_{\alpha}(t)$ and $m_{\alpha}(t)$ of the *solution* of the dynamics $\Pi(t)$. However, just like the relation between $\mathcal{G}(t)$ and $\mathcal{G}^{\mathrm{H}}(t)$, this relation is highly nontrivial whenever $[\mathcal{G}(t), \Pi(t)] \neq 0$. As a result, anti-causal divisibility cannot be easily related to causal divisibility. The fermionic duality for jump operators solves this nontrivial problem by relating the two jump operator sums, as we will explain below.

### 4.2.2 Fermionic sum rule for jump operators

To derive the duality relation for the jump operator sum, we first note a special implication of Eq. (49), the exact fermion-parity zero mode of $\mathcal{G}(t)$, Eq. (53). This is equivalent to a fundamental sum rule for the jump operators:

$$\sum_{\alpha} j_{\alpha}(t)\left[J_{\alpha}(t)^{\dagger}J_{\alpha}(t) - (-1)^{N_{\alpha}}J_{\alpha}(t)J_{\alpha}(t)^{\dagger}\right] = \Gamma\mathbb{1}. \tag{59}$$

This is remarkable since in general the jump operators are not constrained by *any* sum rule independent of model details, and here the only such detail is the lump sum $\Gamma$. Taking the trace, we find that the time-dependent coefficients $j_{\alpha}(t)$ of the *odd-parity* jump operators sum to a constant,

$$\sum_{\alpha} j_{\alpha}(t)\tfrac{1}{2}[1 - (-1)^{N_{\alpha}}] = \tfrac{1}{2}d\Gamma, \tag{60}$$

leaving the *even parity* jump coefficients unrestricted. Although Eqs. (59) and (60) are clearly analogous to the additional sum rules (40)–(41) and originate from the same fermionic duality

---

[16]If $\mathcal{G}^{\mathrm{H}}(t)$ satisfies this condition, then it implies that $\Pi(t) = \Pi_{\mathrm{a}}(t, 0)$ is CP.

relation they are *not* simple consequences of each other. In the resonant level model there are only odd-parity jump operators [Eqs. (11) and (18a)] and the fermionic sum rule for jump operators (59) is obeyed, $\sum_\eta j_\eta(t)[d_\eta^\dagger d_\eta + d_\eta d_\eta^\dagger] = \Gamma\mathbb{1}$, which in this case is a multiple of the scalar sum rule (60), $\sum_\eta j_\eta(t) = \Gamma$.

### 4.2.3 Cross-relations between Heisenberg and Schrödinger jump operators

Inserting Eq. (57) into Eq. (49) and using the fermionic sum rule (59) we obtain[17] $\mathcal{G}^{\mathrm{H}}(t)$ in the form (58) where the effective Heisenberg Hamiltonian equals minus the Schrödinger one evaluated at dual parameters,

$$H^{\mathrm{H}}(t) = -\bar{H}(t). \tag{61a}$$

We thus explicitly recover the closed-system fermionic duality (2) extended nontrivially by the inclusion of the time-dependent renormalization by the environment ($H(t) \neq H$). In close analogy to the measurement-operator duality (39), the Heisenberg jump operators are related pairwise to Schrödinger jump operators at dual parameters:

$$J_\alpha^{\mathrm{H}}(t) = \bar{J}_{\alpha'}(t), \tag{61b}$$

whereas their corresponding coefficients obey

$$j_\alpha^{\mathrm{H}}(t) = (-1)^{N_{\alpha'}} \bar{j}_{\alpha'}(t). \tag{61c}$$

This duality relation implies that the distinct anti-causal divisibility of the dynamics (all $j_\alpha^{\mathrm{H}}(t) > 0$) can be *decided* by the *parameter dependence* of the coefficients determining the causal divisibility properties (all $j_\alpha(t) > 0$). For the resonant level model, this is achieved by simply replotting Fig. 2(a) in units of temperature while varying the coupling $\Gamma$ as shown in Fig. 2(b). The continuation of the causal divisibility boundary to negative coupling $\Gamma$ precisely gives the anti-causal divisibility boundary that was shown in Fig. 2(a).

The duality relation (61c) tells us precisely when causal and anti-causal divisibility coincide: $j_\alpha(t) \geq 0$ for all $\alpha$ must imply $(-\mathbb{1})^{N_\alpha} \bar{j}_\alpha \geq 0$ and vice versa. This imposes a very strong constraint on the parameter dependence of the dynamics. This always holds when the evolution commutes with its generator, which includes the case of Markovian semigroup evolutions. We then have $\mathcal{G}^{\mathrm{H}}(t) = \mathcal{G}(t)^\dagger$, implying by Eq. (22) that $H^{\mathrm{H}}(t) = H(t)$, $J_\alpha^{\mathrm{H}}(t) = J_\alpha(t)^\dagger$ and $j_\alpha^{\mathrm{H}}(t) = j_\alpha(t)$. Moreover, Eq. (61a) becomes $\bar{H}(t) = -H(t)$: the effective Hamiltonian is constrained to change sign under the duality mapping. In this case equation (58) additionally strengthens to a cross relation between the jump operators of $\mathcal{G}(t)$ alone: $J_\alpha(t)^\dagger = \bar{J}_{\alpha'}(t)$ and their coefficients $j_\alpha(t) = (-1)^{N_{\alpha'}} \bar{j}_{\alpha'}(t)$. This extends the results of Ref. [16] for the weak-coupling generators $\mathcal{G}$ in Lindblad form (57).

Beyond this trivial case the two types of divisibility need not coincide, as evidenced by the resonant level model [Fig. 2(a)]. Thus, fermionic duality strongly suggests that anti-causal divisibility generically *differs* from causal divisibility by an explicit strong constraint on model parameters. This is in line with the general intuition that this type of divisibility additionally requires weak causal ordering of the dynamics and is thus a more fragile property. This motivates further investigation, for example in relation to recent work on causal ordering in quantum information theory [66, 67].

We stress that although the duality relations (39) and (61) have a common origin, for general dynamics one cannot derive the jump-operator duality by using the trick of "linearizing" the measurement operators in Eq. (39) as it is possible in the Markovian semigroup limit [68]. The analogy between (39) and (61) is best seen in the Choi-Jamiołkowski correspondence to $\mathcal{G}(t)$ (instead of $\Pi(t)$) as discussed in App. C.

---

[17]In Eq. (49) the parity transformation $\mathcal{P} \bullet \mathcal{P}$ inverts the sign of the odd parity jump coefficients in Eq. (57). Combined with the $\Gamma$-shift in Eq. (49) it transforms the trace-preserving property of $\mathcal{G}(t)$ into the unit-preserving property of $\mathcal{G}^{\mathrm{H}}(t)$ as it should.

#### 4.2.4 Unphysicality of the duality mapping

To conclude we verify that $\bar{\mathcal{G}}(t)$ is the generator of the dual evolution $\bar{\Pi}(t)$: using Eq. (23) and (49) we find

$$\frac{d}{dt}\bar{\Pi}(t) = -i\bar{\mathcal{G}}(t)\bar{\Pi}(t). \tag{62}$$

What is interesting here is that $\bar{\mathcal{G}}(t)$ generates $\bar{\Pi}(t)$ in the same causal order as the Schrödinger evolution $\Pi(t)$. On the other hand the dual propagator $\bar{\Pi}(t)$ is related to the propagator in the Heisenberg picture $\Pi^{\mathrm{H}}(t) = e^{-\Gamma t}\mathcal{P}\bar{\Pi}(t)\mathcal{P}$. This implies that $\bar{\mathcal{G}}(t)$ is related to the generator $\mathcal{G}^{\mathrm{H}}(t)$ acting from the left in the Heisenberg evolution [Eq. (46)] and *not* to $\mathcal{G}(t)^\dagger$ acting from the right. This explains why in Eq. (61c) the jump-coefficients $\bar{j}_\alpha(t)$ are related to the coefficients $j_\alpha^{\mathrm{H}}(t)$ characterizing the *anti-causal* divisibility of the evolution [Eq. (58)] and *not* to the $j_\alpha(t)$ describing ordinary *causal* divisibility. Note that the generator $\mathcal{G}^{\mathrm{H}}$ builds up the Heisenberg evolution in *anti-causal* order in contrast to $\mathcal{G}^\dagger$. These observations are gratifying since they tie the physical divisibility properties of the dynamics to a key step in the derivation of the duality (23), the formal reversal of the causal ordering [Eq. (S-71) of Ref. [10]], within a completely different formalism.

We also observe that the relation between $\Pi^{\mathrm{H}}(t)$ and $\bar{\Pi}(t)$ is reflected by their generators appearing in the duality (49). Written as jump operator sum similar to Eq. (58), the dual generator reads

$$-i\bar{\mathcal{G}}(t) = -i[\bar{H}(t), \bullet] + \sum_\alpha \bar{j}_\alpha(t)\bar{\mathcal{D}}_\alpha(t). \tag{63}$$

The TP property of $\bar{\Pi}(t)$ corresponds to $\mathrm{Tr}\,\bar{\mathcal{G}}(t) = 0$ which is ensured by Eq. (53). In Eq. (63) this property is ensured by the causal structure of the dual dissipators $\bar{\mathcal{D}}_\alpha(t) = \bar{J}_\alpha(t) \bullet \bar{J}_\alpha(t)^\dagger - \frac{1}{2}\{\bar{J}_\alpha(t)^\dagger \bar{J}_\alpha(t), \bullet\}$ which *differs* from the Heisenberg dissipators $\mathcal{D}_\alpha^{\mathrm{H}}(t)$ by the position of the adjoint in the anticommutator [cf. Eq. (58) ff.].

Even though $\bar{\mathcal{G}}(t)$ has the causal structure and the TP property of a Schrödinger picture generator, we know that the dynamics $\bar{\Pi}(t)$ it generates is never CP [Sec. 3.2.3]. This general conclusion is not readily seen from the jump expansion (57) of $\bar{\mathcal{G}}(t)$, which is tailored to reflect divisibility properties.[18] However, it can be seen in the special case where $\mathcal{G}$ is time-constant: then $\Pi(t) = e^{-i\mathcal{G}t}$ is CP-TP if *and only if* $j_\alpha \geq 0$. Since in this case we also have $j_\alpha = (-1)^{N_{\alpha'}}\bar{j}_{\alpha'}$ [Eq. (61) ff.] this implies $\bar{j}_\alpha < 0$ for all odd-parity jump-operators and thus the generated map $\bar{\Pi}(t) = e^{-i\bar{\mathcal{G}}t}$ is not CP.

### 4.3 Time-nonlocal quantum master equation

We now turn to the expression of fermionic duality in the last approach discussed in this paper, which will be particularly important for the application in Sec. 5. We now exploit that the evolution $\Pi(t)$ is also the solution of the completely different time-nonlocal QME

$$\frac{d}{dt}\Pi(t) = -i\int_0^t dt'\mathcal{K}(t-t')\Pi(t'). \tag{64}$$

In contrast to the time-local QME (45), its convolution structure matches the one obtained in the microscopic derivation of the evolution [10–14, 18, 19]: the propagator decomposes into a geometric series of convolutions of memory-kernel blocks $-i\mathcal{K}(t-t')$ of duration $t-t'$, giving a self-consistent Dyson equation: denoting $(f * g)(t) = \int_0^t dt' f(t-t')g(t')$,

$$\Pi = \mathcal{I} + \mathcal{I} * (-i\mathcal{K}) * \mathcal{I} + \mathcal{I} * (-i\mathcal{K}) * \mathcal{I} * (-i\mathcal{K}) * \mathcal{I} + \ldots = \mathcal{I} + \mathcal{I} * (-i\mathcal{K}) * \Pi. \tag{65}$$

---

[18]For time-dependent generators which commute with the evolution, $[\mathcal{G}(t), \Pi(t)] = 0$, we still have a direct relation $j_\alpha(t) = (-1)^{N_{\alpha'}}\bar{j}_{\alpha'}(t)$. If $\Pi(t)$ is CP-divisible, i.e., $j_\alpha(t) \geq 0$ for all $t \geq 0$ then $\bar{j}_\alpha(t) \leq 0$ for all odd-parity operators. As mentioned in footnote 15 this does not allow to infer whether $\bar{\Pi}(t)$ is CP or not. If $\Pi(t)$ is not CP-divisible, the signs of $j_\alpha(t)$ are unrestricted and we cannot conclude either.

Taking the time derivative gives Eq. (64) [Eq. (15)].

By definition we included the time-local closed-system dynamics $L = [H, \bullet]$ into the memory kernel $\mathcal{K}(t) = L\,\delta(t) + \mathcal{K}'(t)$ with the normalization $\int_0^t ds\,\delta(s) = 1$. Using the adjoint of Eq. (64) and (65) we obtain the time-nonlocal Heisenberg QME

$$\frac{d}{dt}\Pi(t)^\dagger = i\int_0^t dt'\mathcal{K}(t-t')^\dagger\,\Pi(t')^\dagger\,, \tag{66}$$

noting that under the convolution one may commute[19] $\Pi(t')$ and $\mathcal{K}(t-t')$.

Inserting the propagator duality (23) and Eq. (64) on the left-hand-side of Eq. (66) we obtain the fermionic duality for the memory kernel

$$\left[\mathcal{K}(t)\right]^\dagger = i\Gamma\mathcal{I}\,\delta(t) - e^{-\Gamma t}\mathcal{P}\bar{\mathcal{K}}(t)\mathcal{P}\,. \tag{67}$$

For the resonant level model the memory kernel (16) indeed obeys this relation: this follows from the parameter dependence of the nontrivial function $\bar{\mathsf{k}}(t) = -\mathsf{k}(t)$ [Eq. (26)] and the parity transformation of the dissipator $\mathcal{P}\mathcal{D}_\eta^\dagger\mathcal{P} = -\mathcal{D}_{-\eta} - \mathcal{I}$.

### 4.3.1 Complex-frequency representation of dynamics

An advantage of the time-nonlocal QME (64) is that it allows a particularly simple explicit expression of $\Pi(t)$ in terms of the Laplace transform $\hat{\mathcal{K}}(\omega) := \int_0^\infty dt\,e^{i\omega t}\mathcal{K}(t)$ of the memory kernel which facilitates further analysis:

$$\hat{\Pi}(\omega) = \frac{i}{\omega - \hat{\mathcal{K}}(\omega)}\,. \tag{68}$$

Laplace transforming relation (23) gives the fermionic duality in frequency-domain reported in Ref. [10]:

$$\hat{\Pi}(\omega)^\dagger = \mathcal{P}\,\hat{\bar{\Pi}}(i\Gamma - \omega^*)\,\mathcal{P}\,. \tag{69}$$

This relates $\hat{\Pi}(\omega)$ and $\hat{\bar{\Pi}}(\omega)$ in complex-frequency regions where either both their Laplace transforms converge, or in regions where both are defined by analytical continuation. The mapping of the complex frequency argument reverses the real energy part of $\omega$, while maintaining the sign of the dissipative imaginary part of $\omega$ up to a shift $i\Gamma$ into the upper half plane. The fermionic duality for the frequency-domain memory kernel has the same structure:[20]

$$\hat{\mathcal{K}}(\omega)^\dagger = i\Gamma\mathcal{I} - \mathcal{P}\hat{\bar{\mathcal{K}}}(i\Gamma - \omega^*)\mathcal{P}\,. \tag{70}$$

### 4.3.2 Unphysicality of the duality mapping

While the above discussed operational approaches concern algebraic properties at each instance of time, the analytical structure of the memory kernel makes explicit how physical properties *evolve* in time. This makes fermionic duality in the frequency domain of independent interest (see below). It also reveals another way in which the dual propagator is unphysical as follows. Since a physical evolution $\Pi(t)$ in general shows oscillations and decay or a combination thereof, its Laplace transform converges only for complex frequencies in the upper half plane. The obtained function $\hat{\Pi}(\omega)$ has a unique extension to the lower half plane where in

---

[19]The identity $\mathcal{K} * \Pi = \Pi * \mathcal{K}$ follows from the two ways of writing the Dyson equation, $\Pi = \mathcal{I} + \mathcal{I} * (-i\mathcal{K}) * \Pi = \mathcal{I} + \Pi * (-i\mathcal{K}) * \mathcal{I} = \mathcal{I} + \mathcal{I} * [\Pi * (-i\mathcal{K})]$ and taking the time derivative.

[20]In the weak coupling limit $\Gamma$ enters in $\hat{\mathcal{K}}(\omega)$ only as a prefactor, $\hat{\mathcal{K}}(\omega) \propto \Gamma$. In this case it is possible to consider a *physical* dual system without inversion of the coupling $\Gamma$ by directly including the sign change of $\hat{\mathcal{K}}$ in a modified duality relation [10, 16, 17].

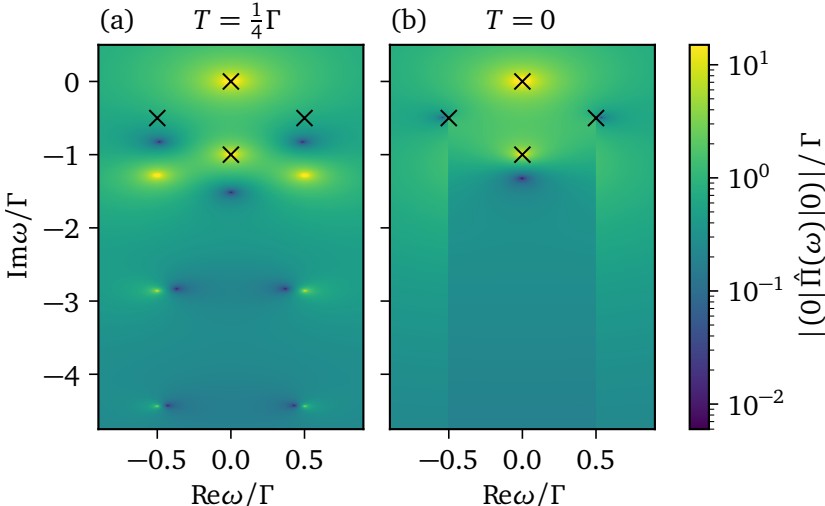

Figure 3: Resonant level model for strong coupling and detuning $\varepsilon - \mu = \Gamma/2$. Plotted is the modulus of the complex valued matrix element $(0|\hat{\Pi}(\omega)|0)$ in units of $\Gamma$ in the complex frequency plane where $|0) = |0\rangle\langle 0|$ denotes the unoccupied state. (a) Finite temperature $T = \Gamma/4$: distinct from the two infinite sets of equidistant poles there are four poles (marked $\times$) at $\omega = 0$, $\omega = -i\Gamma$ and $\omega = \pm\varepsilon - i\Gamma/2$. The last two poles are not visible here but appear in other matrix elements. (b) For $T \to 0$ two branch cuts develop from the sets of equidistant poles. At resonance ($\varepsilon - \mu \ll T$) these poles (branch cuts) cancel exactly leaving just a single pole at $-i\Gamma$ whereas off-resonance ($\varepsilon - \mu \gg T, \Gamma$) they move to the sides where they become suppressed in amplitude. Only in these two limits four poles remain and the dynamics is a semigroup [23]. For $T \to \infty$ the first case always applies.

general it has both poles and branch points. This is illustrated for the resonant level model in Fig. 3. Integrating along any clockwise oriented contour $\mathcal{C}$ enclosing the poles and branch cuts (parallel to the imaginary axis) gives the general solution for the real-time evolution [11, 12]:

$$\Pi(t) = \int_{\mathcal{C}} \frac{d\omega}{2\pi} e^{-i\omega t} \hat{\Pi}(\omega) = -i \sum_p \text{Res}\left[\hat{\Pi}(\omega_p) e^{-i\omega_p t}\right] + \int_{\text{b.c}} \frac{d\omega}{2\pi} e^{-i\omega t} \hat{\Pi}(\omega), \qquad (71)$$

where $\text{Res} f(\omega_p)$ denotes the residue of $f(\omega)$ at $\omega = \omega_p$. In view of our later application in Sec. 5.2, we note that the first term on the right hand side of (71) sums up contributions from two types of poles: those that arise due to the frequency-dependent eigenvalues $\hat{\pi}_i(\omega) = i/[\omega - \hat{k}_i(\omega)]$ obeying the pole equation $\hat{k}_i(\omega_p) = \omega_p$ where $\hat{k}_i(\omega)$ is an eigenvalue of $\hat{\mathcal{K}}(\omega)$, and the remaining poles which also involve the eigenvectors.

Since the parameter map $\Pi(t) \to \bar{\Pi}(t)$ appearing in the duality relation (23) inverts the sign of dissipative decay rates, the Laplace transform of $\bar{\Pi}(t)$ converges only for frequencies *above an imaginary cutoff* in the upper half of the complex plane which is at least $i\Gamma$, the fundamental parity eigenvalue: If $\Pi(t)$ converges to some stationary value $\Pi(\infty)$ this implies by Eq. (23) that $\bar{\Pi}(t)$ diverges at least as fast as $e^{\Gamma t}$ which must be suppressed by $e^{i\omega t}$ in the Laplace transform. Thus, also its analytical structure proves clearly that the *time-dependence* of the dual propagator $\bar{\Pi}(t)$ is not physical, complementing the discussion of the algebraic, operational constraints of CP and TP [Sec. 3.2.3] which are *independent* of time, see also [33].

### 4.3.3 Cross-relations frequency-dependent left and right eigenvectors

Laplace transforming $\Pi(t)$ and $\mathcal{K}(t)$, analytically continuing and diagonalizing gives

$$\hat{\Pi}(\omega) = \sum_i \hat{\pi}_i(\omega) \big| \hat{\pi}_i(\omega) \big) \big( \hat{\pi}'_i(\omega) \big| , \qquad \hat{\mathcal{K}}(\omega) = \sum_i \hat{k}_i(\omega) \big| \hat{k}_i(\omega) \big) \big( \hat{k}'_i(\omega) \big| . \tag{72}$$

Inserted into the memory-kernel duality (70) we obtain for the eigenvalues

$$\hat{k}_j(\omega) = \big[ i\Gamma - \bar{\hat{k}}_i(i\Gamma - \omega^*) \big]^* , \tag{73a}$$

with the duality between left and right eigenvectors

$$\big( \hat{k}'_j(\omega) \big| = \big( \bar{\hat{k}}_i(i\Gamma - \omega^*) \big| \mathcal{P} , \qquad\qquad \big| \hat{k}_j(\omega) \big) = \mathcal{P} \big| \bar{\hat{k}}'_i(i\Gamma - \omega^*) \big) . \tag{73b}$$

Due to the simple relation (68) in the frequency domain the eigenvectors coincide, $\big( \hat{\pi}'_i(\omega) \big| = \big( \hat{k}'_i(\omega) \big|$ and $\big| \hat{\pi}_i(\omega) \big) = \big| \hat{k}_i(\omega) \big)$, with eigenvalues $\hat{\pi}_i(\omega) = i/[\omega - \hat{k}_i(\omega)]$:

$$\hat{\pi}_j(\omega) = \bar{\hat{\pi}}_i(i\Gamma - \omega^*)^* , \tag{74}$$

in agreement with Eq. (69). We stress that the frequency-domain fermionic duality relations (73)–(74) are of independent interest: they are *not* trivial consequences of the time-domain relations (29) since the Laplace transformation and diagonalization do *not* commute. The $\omega$-dependent eigenvectors (eigenvalues) of $\hat{\Pi}(\omega)$ are *not* the Laplace transforms of the $t$-dependent eigenvectors (eigenvalues) of $\Pi(t)$.

For the resonant level model, Laplace transforming Eq. (8) gives (App. D of Ref. [23]):

$$\hat{\Pi}(\omega) = \sum_{\eta=\pm} \frac{i}{\omega + \eta\varepsilon + i\frac{\Gamma}{2}} \big| d^\dagger_\eta \big) \big( d^\dagger_\eta \big| + \frac{i}{\omega} \tfrac{1}{2} \big[ \big| \mathbb{1} \big) + \hat{\mathsf{k}}\big(\omega + i\tfrac{\Gamma}{2}\big) \big| (-\mathbb{1})^N \big) \big] \big( \mathbb{1} \big|$$
$$+ \frac{i}{\omega + i\Gamma} \tfrac{1}{2} \big| (-\mathbb{1})^N \big) \big[ \big( (-\mathbb{1})^N \big| - \hat{\mathsf{k}}\big(\omega + i\tfrac{\Gamma}{2}\big) \big( \mathbb{1} \big| \big] . \tag{75}$$

The left and right eigenvectors are indeed cross-related as dictated by Eq. (73b). In particular, the (non)trivial frequency dependence of the left (right) eigenvector for the eigenvalue with pole $\omega = 0$ *necessarily implies* that the right (left) eigenvector for the eigenvalue with pole $\omega = -i\Gamma$ is (non)trivial as well. Thus, also in the frequency domain duality provides fine-grained insight into the location of nontrivial (non-exponential in time) contributions to the dynamics. In particular, the frequency dependence of the eigenvectors through the Laplace transform of $\mathsf{k}(t)$ [Eq. (5)],

$$\hat{\mathsf{k}}(\omega) = \frac{i}{\pi} \sum_{\eta=\pm} \eta \psi \left( \frac{1}{2} - i \frac{\omega + \eta(\varepsilon - \mu)}{2\pi T} \right) , \tag{76}$$

generates infinitely many additional poles at $\omega = \pm(\varepsilon - \mu) - i\Gamma/2 - i\pi T(2n+1)$, $n = 0, 1, \dots$ due to the digamma function $\psi$. For $T \to 0$ the poles merge to form two branch cuts as shown in Fig. 3. In our application in the next section this analytic structure turns out to provide crucial insights.

## 5 Nonperturbative semigroup approximation and initial slip

Finally, we consider an application of fermionic duality where the insights of several of the discussed approaches come together. We consider analytic approximations to the solution of the time-*local* QME (45), $\frac{d}{dt}\Pi(t) = -i\mathcal{G}(t)\Pi(t)$, constructed from the generator $\mathcal{G}(t)$ which

we assume to be exactly known (best case). This equation naturally suggests a *nonperturbative semigroup approximation* which does not rely on any weak-coupling assumption [5,38]:

$$\Pi^{(1)}(t) := e^{-i\mathcal{G}(\infty)t} = \sum_i e^{-ig_i(\infty)t} |g_i(\infty))(g_i'(\infty)|, \tag{77}$$

requiring only that the generator converges to a stationary value $\mathcal{G}(\infty)$, which is diagonalizable. It is not in general clear how accurate this approximation and corrections to it are. We will show that the quality of these approximations can be deeply understood using its exact relation to the corresponding time-*nonlocal* QME (64) and its memory kernel $\mathcal{K}(t)$ combined with fermionic duality. This effort is motivated by two attractive properties of the approximation (77):

(i) For the large class of evolutions which are *CP-divisible in the stationary limit*, i.e., $j_\alpha(\infty) \geq 0$ in Eq. (57), the approximate evolution (77) is *both CP and TP*. This is in general very difficult to achieve for nonperturbative approximations [29–33]. This class includes dynamics which is *not* a trivial semigroup described by a Lindblad equation, which is already the case for the resonant level model (except for $T = \infty$ or $\varepsilon = \mu$, see Sec. 2). It also includes dynamics which is *not* CP-divisible as long as $j_\alpha(t) \leq 0$ occurs only for finite times.

(ii) The approximate evolution (77) converges to the exact stationary state as we demonstrate below, $e^{-i\mathcal{G}(\infty)t} |\pi_0(\infty)) = |\pi_0(\infty)) = |\rho(\infty))$.

## 5.1 Fixed-point relation between generator $\mathcal{G}$ and memory kernel $\mathcal{K}$

To address this problem, we will make use of a recent exact result [5] which shows that the stationary generator obeys the self-consistent equation [App. E]

$$\mathcal{G}(\infty) = \int_0^\infty dt \, \mathcal{K}(t) e^{it\mathcal{G}(\infty)}. \tag{78}$$

Here the superoperator $\mathcal{G}(\infty)$ takes the role of the complex frequency $\omega$ in the Laplace transform of the memory kernel $\mathcal{K}(t)$. Inserting the spectral decompositions, this implies $\hat{\mathcal{K}}[g_i(\infty)] |g_i(\infty)) = g_i(\infty) |g_i(\infty))$, and a careful analysis shows that the *eigenvalues* of the stationary generator $\mathcal{G}(\infty)$ are *eigenvalue-poles*[21] of $\hat{\Pi}(\omega)$, i.e., $\omega_p = \hat{k}_j(\omega_p)$ [Eq. (68)]. Equation (78) thus states that $\mathcal{G}(\infty)$ "samples" the Laplace transform $\hat{\mathcal{K}}(\omega)$ of the memory kernel precisely at complex frequencies given by the eigenvalues of $\mathcal{G}(\infty)$:

$$\mathcal{G}(\infty) = \sum_i \hat{k}_{j_i}(g_i(\infty)) |\hat{k}_{j_i}(g_i(\infty)))(g_i'(\infty)|. \tag{79}$$

Here $\hat{k}_{j_i}(g_i(\infty)) = g_i$ where $j_i$ denotes the index of the eigenvalue $\hat{k}_j(\omega)$ of $\hat{\mathcal{K}}(\omega)$ which equals $g_i$ at frequency $\omega = g_i$. By trace-preservation, the frequency sampling always includes zero, $g_0(\infty) = 0$, and thus $\mathcal{G}(\infty)$ and $\hat{\mathcal{K}}(0)$ have the same stationary state eigenvector, proving property (ii) mentioned above. However, also *nonzero* complex frequencies are sampled. Thus, only the set of eigenvectors $\{|\hat{k}_{j_i}(g_i(\infty)))\}$, collected from *different* superoperators [$\hat{\mathcal{K}}(\omega)$ at different frequencies $\omega = g_i(\infty)$] provides the full set of right eigenvectors $\{|g_i(\infty))\}$ of the single superoperator $\mathcal{G}(\infty)$. This in turn determines the left eigenvectors $\{(g_i'(\infty)|\}$ of $\mathcal{G}(\infty)$ and thus, remarkably, $\mathcal{G}(\infty)$ can be directly constructed from the memory kernel $\hat{\mathcal{K}}(\omega)$ once one knows the eigenvalues $g_i(\infty)$. For the resonant level model this surprising construction was verified explicitly in Ref. [5].

Ref. [5] focused on the implications of relation (78) assuming that one knows $\mathcal{K}$ and aims to compute $\mathcal{G}$. Here we focus in a sense on the converse question: Using a known $\mathcal{G}$ to construct

---

[21]Because we assume that $\mathcal{G}(\infty)$ exists and Eq. (78) holds, $\mathcal{K}(g_i(\infty))$ cannot have poles in its eigenvectors.

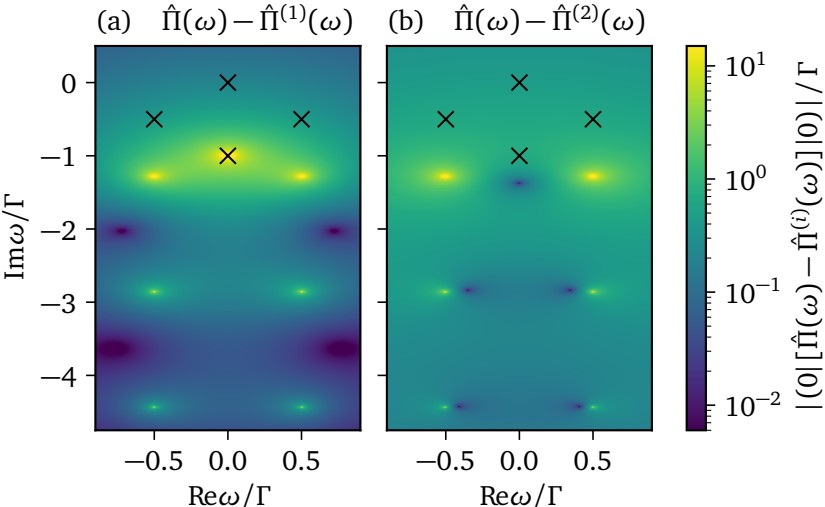

Figure 4: Error analysis for approximations to the resonant level model for coupling $\Gamma = 4T$ and $\varepsilon - \mu = \Gamma/2$. Plotted is the amplitude of one matrix element $(0|[\hat{\Pi}(\omega) - \hat{\Pi}^{(1)}(\omega)]|0)$ (left) and $(0|[\hat{\Pi}(\omega) - \hat{\Pi}^{(2)}(\omega)]|0)$ (right) for the unoccupied state $|0) = |0\rangle\langle 0|$ in the complex frequency plane. Whereas the semigroup approximation $\Pi^{(1)}(t) = e^{-i\mathcal{G}(\infty)t}$ does not fully cancel the pole at $\omega = -i\Gamma$, the initial-slip approximation $\Pi^{(2)}(t) = e^{-i\mathcal{G}(\infty)t}\mathcal{S}$ exactly cancels all four indicated poles of $\hat{\Pi}(\omega)$ in Fig. 3.

an approximation for $\Pi(t)$, what general insights into the quality of the approximation does $\mathcal{K}$ provide, and how does fermionic duality help in this analysis? We consider the simplest approximation beyond the semigroup approximation (77), a *corrected* semigroup which aims at improving the description of the evolution at long times. To illustrate the simplest type of such a correction we assume in the following that the poles of $\hat{\Pi}(\omega)$ at the eigenvalues of $\mathcal{G}(\infty)$ are of first order.[22] We start by noting that by Eq. (79) the eigenvalues of $\mathcal{G}(\infty)$ must be poles in the eigenvalues of $\hat{\Pi}(\omega)$ [first term of Eq. (71)]. Their contribution to the exact evolution can be expressed as

$$\Pi^{(2)}(t) := -i \sum_i e^{-ig_i(\infty)t} \operatorname{Res} \hat{\Pi}(g_i(\infty)) \tag{80a}$$

$$=: e^{-i\mathcal{G}(\infty)t}\mathcal{S}. \tag{80b}$$

Here the sum over $i$ runs over distinct values of $g_i(\infty)$. This indeed looks like an initial slip correction to the semigroup approximation. We will now discuss the quality of the two approximations $\Pi^{(1)}$ and $\Pi^{(2)}$ and then derive an exact restriction on this procedure imposed by fermionic duality.

## 5.2 Nonperturbative semigroup approximation

The quality of the semigroup approximation (77) can be investigated using the result (80a) of the *time-nonlocal* QME approach. We observe that in the complex-frequency domain the error $\hat{\Pi}(\omega) - \hat{\Pi}^{(1)}(\omega)$ has in general the *same* poles as the exact dynamics except for the pole at zero frequency. Thus, the approximation captures only the stationary state systematically.

---

[22]Our assumption is equivalent to $d\hat{k}_{j_i}(\omega)/d\omega|_{\omega=g_i(\infty)} \neq 1$. Higher order poles would require a time-dependent slip superoperator $\mathcal{S}(t) = \mathcal{S}_1 + \mathcal{S}_2 t + \mathcal{S}_3 t^2 + \cdots$ where $\mathcal{S}_n$ is constructed from the residues of all $n^{\text{th}}$ order poles of $\hat{\Pi}(\omega)$ at eigenvalues of $\mathcal{G}(\infty)$.

In Fig. 4(a) this is illustrated for the resonant level model. The reason becomes clear when comparing Eq. (77) with Eq. (80a). Although the decompositions of $\Pi^{(1)}(t)$ and $\Pi^{(2)}(t)$ share the same coefficients and right vectors, only in $\Pi^{(2)}(t)$ the different pole contributions appear with the correct *left* eigenvectors. In Eq. (77) all eigenvalues are exponential functions with prefactor one, while the coefficients in Eq. (80a) may include prefactors from the residuals $\hat{\Pi}(g_i(\infty))$ due to the frequency-dependence of $\hat{\mathcal{K}}(\omega)$ [cf. Eq. (82b)]. Also, the left eigenvectors in Eq. (77) are biorthogonal to the right eigenvectors, while the left eigenvectors of the residuals in Eq. (80a) are collected from kernels $\hat{\mathcal{K}}(\omega)$ at *different frequencies* and do not need to obey such restrictions, as is indeed the case for the resonant level model [5].

### 5.3 Nonperturbative initial slip correction

Within the time-local QME approach one may set up a nonperturbative correction of the semigroup approximation by an initial slip [69–73] using some superoperator $\mathcal{S}$:

$$\Pi^{(2)}(t) = e^{-i\mathcal{G}(\infty)t}\mathcal{S}. \tag{81}$$

This *initial-slip approximation* is precisely what is obtained in the time-*nonlocal* QME approach when selecting the exact poles $g_i(\infty)$ in the contour integration (71) of the inverse Laplace transform. This relation can be used to investigate how well one can do in the time-*local* approach: we know from Eq. (80) that such a correction can be achieved by

$$\mathcal{S} = -i\sum_i \text{Res}\,\hat{\Pi}(g_i(\infty)) \tag{82a}$$

$$= \sum_i \frac{1}{1 - \partial\hat{k}_{j_i}/\partial\omega(g_i(\infty))}\big|g_i(\infty)\big)\big(\hat{k}'_{j_i}(g_i(\infty))\big|. \tag{82b}$$

Note that this expansion is *not* the spectral decomposition of $\mathcal{S}$, since the sets of left and right eigenvectors are not biorthogonal. This is in fact the best one can do for an approximation in the long time limit since in the frequency domain this approximation for the resolvent reads

$$\hat{\Pi}^{(2)}(\omega) = \frac{i}{\omega - \mathcal{G}(\infty)}\mathcal{S} = -i\sum_i \frac{i}{\omega - g_i}\text{Res}\,\hat{\Pi}(g_i), \tag{83}$$

showing that now the exact poles $g_i$ are completely canceled in the error $\hat{\Pi}(\omega) - \hat{\Pi}^{(2)}(\omega)$ as illustrated in Fig. 4(b). Here we assumed that the eigenvalues $g_i(\infty)$ are nondegenerate as is the case in the resonant level model, see App. D for the degenerate case.

In Fig. 5(a) we illustrate that for the resonant level model the inclusion of the exact initial slip $\mathcal{S}$ may indeed lead to a much faster approach to the exact dynamics than the nonperturbative semigroup. In Eq. (80b) this is achieved by first mapping the initial state using $\mathcal{S}$ to a possibly *nonpositive* density operator—reflected in the plot by the unphysical occupation $> 1$—and then applying the semigroup evolution (77). Indeed, although $\mathcal{S}$ and $\Pi^{(2)}(t)$ are both TP maps[23], the map $\mathcal{S}$ is *not* CP. Thus, the initial dynamics is unphysical whenever the slip is nontrivial, since by construction $\Pi^{(2)}(0) = \mathcal{S} \neq \mathcal{I}$ [74–76]. For the same reason, the slip-approximated dynamics (81) is not a semigroup: even though the decay is exponential, any nontrivial slip obstructs addition of the exponentials: $e^{-i\mathcal{G}(\infty)t'}\mathcal{S}e^{-i\mathcal{G}(\infty)t}\mathcal{S} \neq e^{-i\mathcal{G}(\infty)(t+t')}\mathcal{S}$. Although this is not a problem—the exact dynamics $\Pi(t)$ is not a semigroup either—it may be overlooked that summing isolated pole contributions [Eq. (80a)] in the time-*nonlocal* QME approach, one does *not* obtain a Markovian semigroup approximation.

In Fig. 6 we investigate this correction more closely for the resonant level model by plotting the time at which $\Pi^{(2)}(t)$ becomes CP, a necessary indicator of quality. Interestingly, near

---

[23]$\text{Tr}\,\mathcal{S} = \text{Tr}$ since $\text{Tr}\,g_i = (g'_0|g_i) = \delta_{i0}$ and $(\pi'_0| = \text{Tr}$.



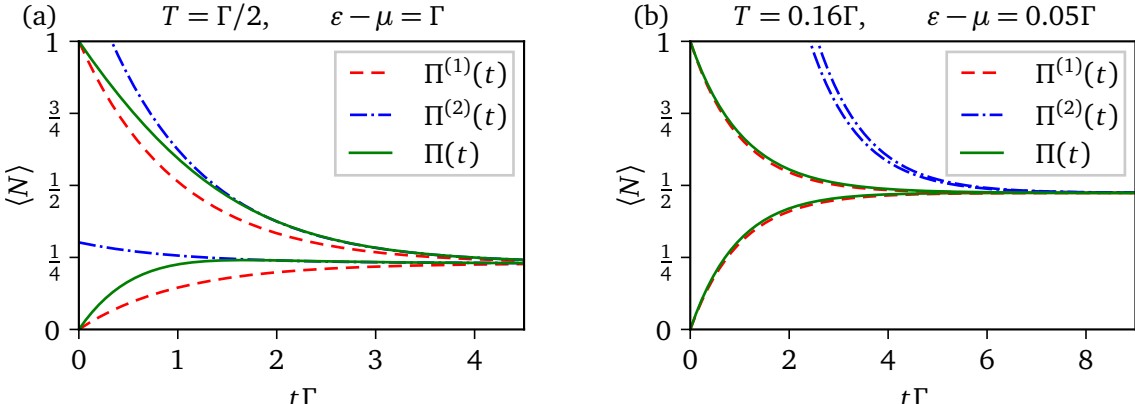

Figure 5: Resonant level model: Nonperturbative semigroup [Eq. (77), dashed] and initial slip approximation [Eq. (80), dash-dotted] compared to the exact occupation dynamics (solid line). (a) For generic level positions $\varepsilon$ and couplings $\Gamma \lesssim 2\pi T$ the slip-corrected dynamics coincides with the exact result well before reaching the stationary value, while the semigroup approximation converges much later. In this case the time-nonlocal QME (64) solved by selection of the poles [Eq. (83)]— automatically including semigroup plus initial slip—is advantageous. (b) Surprisingly, near isolated points in the $\varepsilon, \Gamma$-parameter space the initial slip correction worsens the reliable semigroup approximation (77) based on the time-local QME. The precise positions at which this failure occurs are predicted by fermionic duality: As explained after Eq. (85), they are a consequence of the constraints it imposes on the slippage superoperator. The *increased* error introduced by the slip correction can also be understood as a failure to account for cancellation by eigen*vector* poles responsible for the branch cuts of the dynamics at $T = 0$.

resonance, $\varepsilon = \mu$, this time may *diverge* for specific physical values of the strong coupling. Fig. 5(b) illustrates that in their vicinity the initial slip may give a very large correction even though the semigroup approximation is very close to the exact dynamics. This breakdown is easily overlooked when constructing the slip approximation within the *time-local* formulation in which the frequency-domain structure is not available. This crucial insight is enabled by the fixed-point relation (78) of Ref. [5].

We note that in the time-local approach, one might expect the exact slip superoperator to be expressible as $\mathcal{S} = \lim_{t\to\infty} e^{i\mathcal{G}(\infty)t}\Pi(t)$ but this limit may actually *fail* to exist. In the resonant level model this indeed happens when the coupling exceeds a sharp threshold, $\Gamma \geq 2\pi T$. However, one can regularize this expression to recover Eq. (82) by taking the zero-frequency residual of its Laplace transform $\mathfrak{L}$: $\mathcal{S} = -i \operatorname{Res} \mathfrak{L}\big[e^{i\mathcal{G}(\infty)t}\Pi(t)\big](0)$. We further note that Markovian approximations can also be performed starting from the Heisenberg equation of motion $\frac{d}{dt}A(t) = i\mathcal{G}^{\mathrm{H}}(t)A(t)$ as is often done in quantum optics. However, naively following the same steps in this picture leads to *different results* with problematic features. This further subtlety and its resolution are discussed in App. D.

## 5.4 Fermionic duality for the initial slip

The restrictions imposed by fermionic duality now provide a crucial insight: already for the resonant level model, divergences of the CP-time quality indicator [Fig. 6] are *unavoidable*. This can be understood by taking the stationary limit of the duality relation (49) which, provided

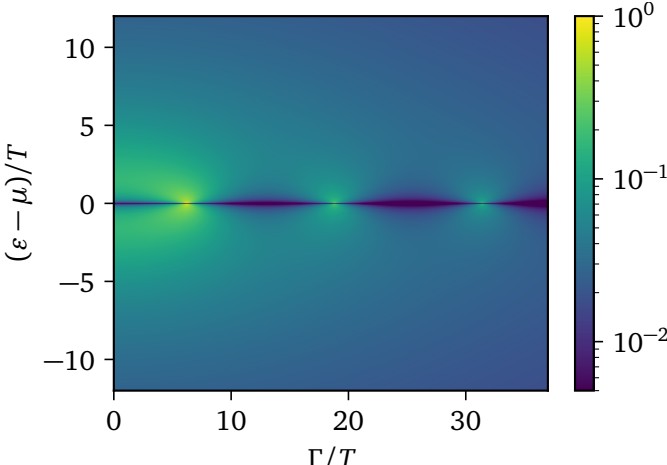

Figure 6: Time after which the initial slip approximation (80b) to the resonant level evolution becomes completely positive, in units of inverse temperature. For $|\varepsilon - \mu| \to \infty$ the semigroup approximation is exact and the slip correction vanishes. Arbitrarily close to the discrete parameter points (87) [Fig. 5(b)] the correction diverges even though the semigroup is very accurate. In between these points the correction remains small [Fig. 5(a)].

$\mathcal{S}^{-1}$ exists, simplifies to

$$\mathcal{G}^{\mathrm{H}}(\infty) = \left[ \mathcal{S}^{-1} \mathcal{G}(\infty) \mathcal{S} \right]^{\dagger} = i\Gamma \mathcal{I} - \mathcal{P} \overline{\mathcal{G}(\infty)} \mathcal{P} \,. \tag{84}$$

Here $\overline{\mathcal{G}(\infty)}$ applies the duality mapping *after* the stationary limit [App. D]. This matters since although the generator $\mathcal{G}(\infty)$ exists for all parameters of the model this is not true for $\mathcal{G}^{\mathrm{H}}(\infty)$ and $\bar{\mathcal{G}}(\infty)$, a further illustration of the nontrivial relation between the Schrödinger and Heisenberg time-local generators. Importantly, the slip superoperator that appears here obeys a separate fermionic duality which is another key result of the paper:

$$\mathcal{S}^{\dagger} = \mathcal{P} \bar{\mathcal{S}} \mathcal{P} \,. \tag{85}$$

For the resonant level model, the exact relations (84) and (85), together with the trace preserving property of $\mathcal{S}$, *completely fix* the relevant part of the slip superoperator constructed from the generator $\mathcal{G}(\infty)$ [App. D]:

$$\mathcal{S} = \mathcal{I} + \tfrac{1}{2} \sum_{\eta} \eta \hat{\mathsf{k}}(\eta i \tfrac{1}{2}\Gamma) \left| (-\mathbb{1})^{N} \right) \left( \mathbb{1} \right| \,, \tag{86}$$

where $\hat{\mathsf{k}}(i\tfrac{1}{2}\Gamma) = (2/\pi) \operatorname{Im} \psi\left( \tfrac{1}{2} + [\Gamma/2 + i(\varepsilon - \mu)]/(2\pi T) \right)$ [cf. Eq. (76)]. As function of parameters the expression $\hat{\mathsf{k}}(i\tfrac{1}{2}\Gamma)$ dictated by fermionic duality is singular at physical parameter points

$$\varepsilon = \mu \pm 0^{+} \,, \qquad\qquad \Gamma = (1 + 2n)2\pi T \,, \tag{87}$$

with $n = 0, 1, 2, \ldots$, causing the slip approximation (80b) to break down as in Fig. 5(b). Moreover, the Heisenberg and dual generator diverge here. This illustrates that fermionic duality can be a useful tool for understanding the often highly nontrivial properties of *time-local* generators of quantum evolutions which complicate analytic approximations.

In the *time-nonlocal* approach the breakdown can be clearly understood in the frequency representation: Close to the values (87), three poles in the exact result for $\hat{\Pi}(\omega)$ [Eq. (75)] approach the same point ($\omega = -i\Gamma$). One pole comes from an eigenvalue of $\hat{\Pi}(\omega)$ (lowest cross

in Fig. 3) and two others come from its eigen*vectors* (top two unmarked poles in Fig. 3). The prefactor of the eigenvalue-pole diverges as function of *parameters* as $1/(\varepsilon - \mu)$ but in the exact result this is canceled by corresponding divergence of the prefactors of the two eigenvector poles. However, in the slip approximation the latter two poles are discarded [Fig. 4], leaving the spurious divergence of the remaining one.

## 6 Discussion

In this paper we have shown that for a large class of fermionic open systems the nontrivial relation between state and observable evolution can be completely bypassed by a simple fermionic duality mapping that exploits and generalizes *the functional dependence* of the two evolutions *on the microscopic physical parameters*. We have shown that this works for essentially all canonical approaches used in quantum transport, open-system dynamics and quantum information theory *without* introducing any assumptions (such as weak coupling, high temperature, various Markovian approximations, etc.) except for wide-band coupling to the reservoirs. The obtained fermionic duality relations are summarized in Table 1. In the superoperator-based approaches these imply exact parity eigenvalues and eigenvectors [Eqs. (32), (53)] and nontrivial cross-relations for the *entire* spectrum [Eqs. (29), (50), (73), (74)]. Correspondingly, in the operational approaches we derived additional fermionic sum rules for measurement and jump operators [Eqs. (40), (59)] and their scalars coefficients [Eqs. (41), (42), (60)], and nontrivial cross-relations for these *entire* sets of operators [Eqs. (39), (61)]. Combining the latter approaches with fermionic duality naturally led us to consider a new type of divisibility of the dynamics: We noted that the Schrödinger and Heisenberg time-local generators through their quantum-jump coefficients encode both ordinary causal and anti-causal divisibility, respectively. Using fermionic duality we showed how the operational condition of anti-causal divisibility can be inferred from the condition of ordinary, causal divisibility. Dynamics which does not commute with its generator may be causally divisible but fail to be anti-causally divisible, as we demonstrated by an explicit example. This provides definite information about the causal ordering of the dynamics, i.e., whether the ordering of dividing the dynamics matters.

Throughout we emphasized both the usefulness of duality relations such as $\Pi(t)^{\dagger} = e^{-\Gamma t} \mathcal{P} \bar{\Pi}(t) \mathcal{P}$ and the unphysicality of mappings such as $\Pi(t) \mapsto \bar{\Pi}(t)$. The usefulness was illustrated by identifying various "symmetries" in the dynamics of the resonant level model which went unnoticed so far. Importantly, our results apply much more generally and are therefore relevant for a large class of outstanding dynamical open-system problems in regimes of strong coupling, strong interactions, finite temperature and nonequilibrium. This holds in particular for much of the analysis and applications in Sec. 5, which apply to complex models of high interest for which the calculations are of course very complicated. Here one should remember that the merit of duality lies in simplifying a calculation given a method of choice, not in providing this method. In this application we exploited fermionic duality to provide deeper insight into a nonperturbative semigroup approximation which respects CP-TP for a broad class of dynamics, using the key result from Ref. [5] that relates time-local and non-local QMEs. We showed that inclusion of the initial-slip correction [Eq. (80b)] in the *time-local* approach (TCL) corresponds precisely to a selection of *poles* in the *time-nonlocal* approach (Nakajima-Zwanzig). We found that even for the resonant level model this correction unexpectedly fails at specific physical parameters and that this failure is unavoidable due to fermionic duality constraints. The failure of the slip-correction is interesting: in the resonant level model it is caused by the poles that form branch-cuts in the limit $T \to 0$. That branch cuts invalidate exponential approximations at $T = 0$ is not unexpected, but here we found that *precursors of branch-cuts* already cause havoc *at finite temperature $T$*.

The unphysicality of the duality mapping appeared particularly clearly in the *operational* formulations tailored to quantum information theory which show that the evolution $\bar{\Pi}(t)$ at dual parameters violates complete positivity. We highlighted this point since it implies that when tacitly assuming this valid and important restriction in any of the operational approaches one may easily overlook the powerful constraints imposed by fermionic duality. For example, in the operational approach evolution maps are invariably written as $\Pi = \sum_\alpha M_\alpha \bullet M_\alpha^\dagger$ (which we avoided doing) by anticipating positive coefficients in the operator sum and absorbing them into the norm of the measurement operators, i.e., $\mathrm{Tr}\, M_\alpha^\dagger M_\alpha = m_\alpha$. This automatically eliminates the dual superoperator $\bar{\Pi}(t)$ with its benefits from further considerations since it has no such expression.

**Relation to other works.** The fermionic duality is most closely related to the extension of PT-symmetry [77,78] to dissipative systems [79,80]. Unlike ordinary symmetries which relate for example the evolution to itself by conjugation with a symmetry transformation, the evolution is related to its adjoint. In the present paper we have emphasized this relation as a connection between the mathematically and physically distinct evolutions of states and observables. In the Refs. [79,80] this was achieved under the strong assumption of Markovianity in the sense of semigroup divisibility (Lindblad). The fermionic duality which was derived in Ref. [10] is instead based on far less restrictive assumptions and is applicable to strongly non-Markovian dynamics such as in the resonant level model.

The involvement of the adjoint sets fermionic duality apart from standard symmetry consideration transposed to Liouville space—called "weak symmetry" in Ref. [81]—where the time-local generator commutes with the symmetry superoperator. Open systems also allow for a notion of "strong symmetry" where the Hamiltonian and jump operators of this time-local generator commute with a symmetry *operator*. This stronger notion introduced in Ref. [81] for semigroup-Markovian systems played a role in our analysis albeit in modified form (the jump operators may either commute or anticommute with the fermion parity operator, see Eq. (57) ff. and App. C). However, fermionic duality is distinct from both these notions of symmetry.

We furthermore note that after the original derivation of fermionic duality in Ref. [10] subsequent work appeared [82,83] which exploited similar tricks to simplify the calculation of open system evolutions, such as unphysical, non-Hermitian coupling to reservoirs with structureless wide bands. However, in addition to focussing on bosonic systems, these works relate the environment of a system of interest to a simpler, effective environment to reduce the computational complexity. In contrast, the fermionic duality relates a system and its environment to an equally complex dual system and environment, in order to exploit the functional parameter dependence of the evolution.

Finally, our finding of a new type of divisibility of the dynamics raises an interesting question regarding stochastic simulations as different types of divisibility are at the basis of different simulation methods. Whereas CP-divisibility of the dynamics enables an implementation that directly uses the jump operators to generate quantum jumps [51,84,85], P-divisibility allows only for an indirect implementation of the jumps via Diosi's rate-operator [51,86,87]. If the evolution has no divisibility property, its simulation is more complicated, requiring reverse quantum jumps connecting trajectories [51]. However, all these distinctions are based on causal division of the dynamics. Fermionic duality surprisingly enables conclusions about anti-causal divisibility, an apparently new concept with a clear operational formulation independent of fermionic duality. It presents a refined distinction between different types of dynamics. For the resonant level model we found that anti-causal CP-divisibility is lost while the dynamics remains causally CP-divisible and thus efficiently simulateable. It is an interesting open question how to detect the loss of this property on the level of simulated quantum

trajectories.

**Outlook.** Exploiting the results reported here in applications to interacting models with strong coupling requires that one uses an approach that *maintains fermionic duality* in approximations. The renormalized perturbation theory [13, 14] which was used originally [10, 16] to derive the duality relation (3) exhibits this feature: It preserves duality order by order in a renormalized temperature-dependent coupling, see Ref. [88] for a recent implementation. Ordinary perturbation theory in the bare coupling $\Gamma$ [89–91] already *breaks* fermionic duality in the next to leading order $\Gamma^2$. For the Anderson model this breakage seems specifically related to the electron pair-tunneling contributions [90, 92, 93].

For calculations involving stronger coupling the renormalization-group approach of Refs. [11–13, 94–99] is well-suited. Its original formulation [11, 13] is build on the renormalized perturbation expansion, which explicitly preserves fermionic duality [10]. Although in Ref. [13] the first implications of fermionic duality were discovered and applied in a very advanced context, it remains an interesting open question which truncation schemes for this exact hierarchy of the RG equations maintain the fermionic duality. Moreover, the potential advantages of the duality for the more recent E-flow formulation [95–97, 99] also remain unexplored. Finally, it is of interest to understand how approximations can be formulated within other nonperturbative approaches in a way that maintains fermionic duality, i.e., by which rules. A key step in this direction would be an elementary microscopic derivation of fermionic-duality *within* these approaches. Our finding that fermionic duality takes a simple form in each of the canonical approaches to quantum dynamics suggests that this is possible.

## Acknowledgements

We thank J. Splettstoesser, M. Pletyukhov and V. Reimer for useful discussions.

**Funding information**   V. B. and K. N. were supported by the Deutsche Forschungsgemeinschaft (RTG 1995). J. S. was supported by the Danish National Research Foundation, the Danish Council for Independent Research | Natural Sciences, and the Microsoft Corporation.

## A   Duality for Choi-Jamiołkowski state of propagator $\Pi(t)$

*CP and CJ operator.* A dynamical map $\rho(t) = \Pi(t)\rho(0)$ is completely positive (CP) if and only if it preserves positivity when evolving the system together a non-evolving auxiliary system: $(\Pi(t) \otimes \mathcal{I})\rho_{\text{ext}}(0) \geq 0$ for any initial state $\rho_{\text{ext}}(0)$ of the system plus any auxiliary system. This is equivalent to positivity for the worst case $\rho_{\text{ext}}(0) = \frac{1}{d}|\mathbb{1}\rangle\langle\mathbb{1}|$ of a maximally entangled state $|\mathbb{1}\rangle = \sum_k |k\rangle|k\rangle$ on the tensor product of the system Hilbert space with an auxiliary copy of itself. Thus, the so-called Choi-Jamiołkowski (CJ) operator should be positive:

$$\text{choi}[\Pi(t)] := \big(\Pi(t) \otimes \mathcal{I}\big)|\mathbb{1}\rangle\langle\mathbb{1}| . \tag{88}$$

The operators $M_\alpha(t)$ in the operator sum (35) for a Hermicity-preserving superoperator $\Pi(t)$ are obtained by diagonalizing the Hermitian choi$[\Pi(t)]$:

$$\text{choi}[\Pi(t)] = \sum_\alpha m_\alpha(t)|M_\alpha(t)\rangle\langle M_\alpha(t)| . \tag{89}$$

When diagonalized with eigenvectors *normalized* to 1 the real eigenvalues $m_\alpha(t)$ are the coefficients in the operator sum (35). If $\Pi(t)$ is CP they are positive since then choi $\Pi(t) \geq 0$ as noted

in the main text. The bipartite eigenvectors $|M_\alpha(t)\rangle = (M_\alpha(t) \otimes \mathbb{1})|\mathbb{1}\rangle = \sum_{ij}\langle i|M_\alpha(t)|j\rangle\,|i\rangle|j\rangle$ determine the matrix elements of the canonical measurement operators relative to the chosen basis $\{|i\rangle\}$.

*Parity of measurement operators.* Fermion superselection for the total system evolution and initial reservoir state implies that the reduced system evolution $\Pi(t)$ commutes with the parity superoperator $(-\mathbb{1})^{L_N} := (-\mathbb{1})^N \bullet (-\mathbb{1})^N$. This implies that $\mathrm{choi}[\Pi(t)]$ commutes with the bipartite parity $(-\mathbb{1})^N \otimes (-\mathbb{1})^N$,

$$\mathrm{choi}[\Pi(t)] = \mathrm{choi}\big[(-\mathbb{1})^{L_N}\Pi(t)(-\mathbb{1})^{L_N}\big] = (-\mathbb{1})^N \otimes (-\mathbb{1})^N\,\mathrm{choi}[\Pi(t)]\,(-\mathbb{1})^N \otimes (-\mathbb{1})^N\,. \quad (90)$$

This shows that the eigenvectors $|M_\alpha(t)\rangle$ have definite bipartite parity and thus the measurement operators must have definite parity: $(-\mathbb{1})^N M_\alpha(t)(-\mathbb{1})^N = (-1)^{N_\alpha} M_\alpha(t)$ with $N_\alpha =$ even or odd, as claimed in the main text.

*Fermionic duality.* Using the property that in the maximally entangled state the action of any operator on the system is perfectly transposed to its copy, $A \otimes \mathbb{1}|\mathbb{1}\rangle = \mathbb{1} \otimes A^T|\mathbb{1}\rangle$, the relation (23) implies the *fermionic duality for the CJ state*

$$\mathrm{choi}[\Pi(t)^\dagger] = \mathbb{S}\big(\mathrm{choi}[\Pi(t)]\big)^*\mathbb{S} = e^{-\Gamma t}\big((-\mathbb{1})^N \otimes (-\mathbb{1})^N\big)\mathrm{choi}[\bar{\Pi}(t)]\,, \quad (91)$$

involving the bipartite swap operator $\mathbb{S}|i\rangle|j\rangle = |j\rangle|i\rangle$. Thus, if $|M_{\alpha'}(t)\rangle$ is a *right* eigenvector of $\mathrm{choi}[\Pi(t)]$ with eigenvalue $m_{\alpha'}(t)$, then $[\mathbb{S}|\bar{M}_{\alpha'}(t)\rangle]^* = |\bar{M}_{\alpha'}(t)^\dagger\rangle$ is *also* a *right* eigenvector with eigenvalue $m_\alpha(t) = e^{-\Gamma t}(-1)^{N_{\alpha'}}\bar{m}_{\alpha'}(t)$ proving Eq. (39b) in the main text. Using $|M_\alpha\rangle = (M_\alpha \otimes \mathbb{1})|\mathbb{1}\rangle$ we also establish the fermionic duality (39a) for measurement operators.

*Degenerate coefficients $m_\alpha(t)$.* If some coefficient $m_\alpha(t)$ is a degenerate eigenvalue of $\mathrm{choi}[\Pi(t)]$ with eigenvectors denoted $\{|M_{\alpha\lambda}\rangle\}_{\lambda\in\alpha}$, then the above argument establishes a correspondence between Hermitian eigenprojectors, $P_\alpha \equiv \sum_{\lambda\in\alpha}|M_{\alpha\lambda}\rangle\langle M_{\alpha\lambda}|$. The eigenvalues $m'_\alpha$ and $m_\alpha(t) = e^{-\Gamma t}(-1)^{N_{\alpha'}}\bar{m}_{\alpha'}(t)$ are *equally degenerate* and the projectors on their eigenspaces are related by

$$(\mathbb{S}P_\alpha\mathbb{S})^* = \bar{P}_{\alpha'}\,. \quad (92)$$

This means that corresponding *partial operator sums* for the dual eigenvalue pair $\alpha$ and $\alpha'$ are equal:

$$\sum_{\lambda\in\alpha}M^\dagger_{\alpha\lambda} \bullet M_{\alpha\lambda} = \sum_{\lambda\in\alpha'}\bar{M}_{\alpha'\lambda} \bullet \bar{M}^\dagger_{\alpha'\lambda}\,. \quad (93)$$

## B  Relation specific to resonant level model

In addition to the generally valid duality relation (49), the time-local generator of the resonant level model obeys another, simpler relation. This may be understood also from the formal similarity of Eq. (8) and Eq. (18). Despite the fact that $\Pi(t) = \mathcal{T}\exp\big(-i\int_0^t ds\mathcal{G}(t)\big)$ and the time-ordering $\mathcal{T}$ is nontrivial for this model, it holds true that $\Pi(t) = \exp(-it\mathcal{G}(t))|_{\mathsf{g}(t)\to\mathsf{p}(t)}$. Inserting this into the relation (23) and using $\bar{\mathsf{p}}(t) = -\mathsf{p}(t)$ one obtains by comparing exponents

$$\big[\mathcal{G}(t)\big]^\dagger = i\Gamma\mathcal{I} + \mathcal{P}\,\mathcal{G}(t)|_{\mathsf{g}(t)\to-\mathsf{g}(t)}\,\mathcal{P}\,, \quad (94)$$

where on the right we replace "by hand" $\mathsf{g}(t) \to -\mathsf{g}(t)$ in analogy to the transformation of $\mathsf{p}(t)$ under the parameter substitution. As a result, the left and right eigen*vectors* of $\mathcal{G}(t)$ are formally related by taking the adjoint, multiplying with the fermion parity $(-\mathbb{1})^N$ and replacing $\mathsf{g}(t) \to -\mathsf{g}(t)$, as one can verify in Table 2. Although relation (94) is simpler and inferred by inspection, it is *not* valid for the general class of models for which Eq. (50) holds.

# C  Duality for Choi-Jamiołkowski operator of generator $\mathcal{G}(t)$

*Jump expansion for $\mathcal{G}(t)$.* The canonical jump-expansion for a time-local generator $\mathcal{G}(t)$ is obtained by constructing its CJ-operator (88) proceeding analogous to $\Pi(t)$ as in App. A. However, $\mathcal{G}(t)$ is *not* a CP map (unlike $\Pi(t)$) and we can only use that $\operatorname{Tr}\mathcal{G} = 0$ (instead of $\operatorname{Tr}\Pi = \operatorname{Tr}$), and that $-i\mathcal{G}$ is Hermicity-preserving and thus $\operatorname{choi}[-i\mathcal{G}(t)]$ is Hermitian. The canonical form

$$\operatorname{choi}[-i\mathcal{G}(t)] = |\mathbb{1}\rangle\langle B| + |B\rangle\langle\mathbb{1}| + \sum_\alpha j_\alpha |J_\alpha\rangle\langle J_\alpha| \tag{95}$$

is obtained by diagonalizing the projection of $\operatorname{choi}[-i\mathcal{G}(t)]$ on the orthogonal space of the maximally entangled state $|\mathbb{1}\rangle := \sum_k |k\rangle|k\rangle$. This gives the last term in Eq. (95) and the first two terms account for the remaining matrix elements. The projector to this orthogonal space is denoted as $Q := \mathbb{1} - |\mathbb{1}\rangle\langle\mathbb{1}|/d$. Splitting $B = \operatorname{Re}B + i\operatorname{Im}B$ one checks that the Hermitian part is fixed to $\operatorname{Re}B = -\frac{1}{2}\sum_\alpha j_\alpha J_\alpha^\dagger J_\alpha$ by the condition $\operatorname{Tr}\mathcal{G}(t) = 0$. The remaining anti-Hermitian part defines the effective Hamiltonian $H(t) = -\operatorname{Im}B(t)$ in Eq. (57).

*Parity of jump operators.* In our case we can also use that fermion superselection, $[(-\mathbb{1})^{L_N}, \Pi(t)] = 0$, implies $[(-\mathbb{1})^{L_N}, \mathcal{G}(t)] = 0$. Thus, $\operatorname{choi}[\mathcal{G}(t)]$ also commutes with the bipartite parity since Eq. (90) also applies with $\Pi \to \mathcal{G}$. Since $|\mathbb{1}\rangle$ has even bipartite parity, we have $(-\mathbb{1})^N \otimes (-\mathbb{1})^N|B\rangle = |B\rangle$ or $[B, (-\mathbb{1})^N] = 0$. This shows that the Hamiltonian part in Eq. (57) commutes with fermion parity $[H(t), (-\mathbb{1})^N] = 0$.

The remaining eigenvectors of the projection with $Q$ have definite bipartite parity of either sign, $(-\mathbb{1})^N \otimes (-\mathbb{1})^N|J_\alpha\rangle = (-1)^{N_\alpha}|J_\alpha\rangle$ for $N_\alpha$ being even or odd. These determine the jump operators through $|J_\alpha(t)\rangle = (J_\alpha(t) \otimes \mathbb{1})|\mathbb{1}\rangle$ for which $(-\mathbb{1})^N J_\alpha(t)(-\mathbb{1})^N = (-1)^{N_\alpha}J_\alpha(t)$ as claimed in the main text. Note that this implies that $\operatorname{Re}B$ has even parity consistent with the above.

*Degenerate coefficients $j_\alpha(t)$.* If some coefficient $j_\alpha$ is a degenerate eigenvalue of $Q\operatorname{choi}[-i\mathcal{G}(t)]Q$ in the construction of Eq. (95) then similar remarks apply as for the measurement operators. The partial operator sums for the dual eigenvalue pair $\alpha$ and $\alpha'$ are equal,

$$\sum_{\lambda \in \alpha'} J_{\alpha'\lambda}^{\mathrm{H}} \bullet J_{\alpha'\lambda}^{\mathrm{H}}{}^\dagger = \sum_{\lambda \in \alpha} \bar{J}_{\alpha\lambda} \bullet \bar{J}_{\alpha\lambda}^\dagger, \tag{96}$$

instead of the individual jump operators [Eq. (61b)].

*Evolution commutes with its generator.* In the special case where $[\mathcal{G}(t), \Pi(t)] = 0$ things simplify, $i\mathcal{G}^{\mathrm{H}}(t) = [-i\mathcal{G}(t)]^\dagger$, and we have by Eq. (22) $H^{\mathrm{H}}(t) = H(t)$, $J_\alpha^{\mathrm{H}}(t) = J_\alpha(t)^\dagger$ and $j_\alpha^{\mathrm{H}}(t) = j_\alpha(t)$ in Eq. (58). Comparing the latter with the jump expansion for the right hand side of Eq. (49),

$$\begin{aligned}
\operatorname{choi}[i\mathcal{G}(t)^\dagger] &= \operatorname{choi}\left[-\Gamma\mathcal{I} - i\mathcal{P}\bar{\mathcal{G}}(t)\mathcal{P}\right] \\
&= |\mathbb{1}\rangle\left[\langle\bar{B}| - \tfrac{\Gamma}{2}\langle\mathbb{1}|\right] + \left[|\bar{B}\rangle - \tfrac{\Gamma}{2}|\mathbb{1}\rangle\right]\langle\mathbb{1}| + \sum_\alpha (-1)^{N_\alpha}\bar{j}_\alpha|\bar{J}_\alpha\rangle\langle\bar{J}_\alpha|,
\end{aligned} \tag{97}$$

we find the result of the main text $\bar{H}(t) = -H(t)$, $J_\alpha(t)^\dagger = \bar{J}_{\alpha'}(t)$ and $j_\alpha(t) = (-1)^{N_{\alpha'}}\bar{j}_{\alpha'}(t)$ for nondegenerate coefficients.

# D  Duality for stationary generator and slip superoperator

*Duality for the slip superoperator $\mathcal{S}$ [Eq. (86)].*  Using the $t \rightarrow \infty$ limit of Eq. (50a), $g_j(\infty) = \left[i\Gamma - \bar{g}_i(\infty)\right]^*$, and Eq. (69) in Eq. (82) we obtain writing $g_i$ for $g_i(\infty)$

$$\mathcal{P}\bar{\mathcal{S}}\mathcal{P} = -i\sum_i \mathcal{P}\operatorname{Res}\hat{\bar{\Pi}}(\bar{g}_i)\mathcal{P} = -i\sum_i \operatorname{Res}\left[\hat{\Pi}(-i\Gamma - \bar{g}_i^*)^\dagger\right] \tag{98a}$$

$$= i\sum_i \left[\operatorname{Res}\hat{\Pi}(-i\Gamma - \bar{g}_i^*)\right]^\dagger = \sum_j \left[-i\operatorname{Res}\hat{\Pi}(g_j)\right]^\dagger = \mathcal{S}^\dagger. \tag{98b}$$

*Initial slip $\mathcal{S}$ for resonant level model.*  The nontrivial part of the slip $\Delta := \mathcal{S} - \mathcal{I}$ obeys three equations by the fact that $\mathcal{S}$ is TP, Eq. (85) and Eq. (84),

$$\left(\mathbb{1}\middle|\Delta = 0, \qquad \Delta^\dagger = \mathcal{P}\bar{\Delta}\mathcal{P}, \qquad \mathcal{G}(\infty)(1 + \Delta) = (1 + \Delta)[-i\Gamma\mathcal{I} - \mathcal{P}\overline{\mathcal{G}(\infty)}^\dagger\mathcal{P}]. \tag{99}$$

Solving these equations for the resonant level model gives Eq. (86) up to an additional term $\sum_\eta \alpha_\eta(t)\middle|d_\eta\big)\big(d_\eta\middle|$ with undetermined function $\alpha_\eta(t)^* = \bar{\alpha}_\eta(t)$. The latter term is irrelevant for the occupation dynamics, in which the breakdown of the initial slip correction occurs. Thus duality fixes the relevant part. Note that the evolution of the coherences is already exact in the semigroup approximation. Setting $\alpha_\eta(t) = 0$ we obtain the exact result.

*Stationary limit of time-local duality (49) [Eq. (84)].*  Noting that the left action of $\mathcal{G}(\infty)$ on the slip operator gives $\mathcal{G}(\infty)\mathcal{S} = -i\sum_i g_i \operatorname{Res}\hat{\Pi}(g_i)$ by Eq. (82b), we insert the dual parameters and follow the same steps as in Eq. (98):

$$\mathcal{P}\overline{\mathcal{G}(\infty)}\bar{\mathcal{S}}\mathcal{P} = -i\sum_i \bar{g}_i\mathcal{P}\operatorname{Res}\hat{\bar{\Pi}}(\bar{g}_i)\mathcal{P}$$

$$= i\sum_j (i\Gamma - g_j^*)\left[\operatorname{Res}\hat{\Pi}(g_j)\right]^\dagger = \{[-i\Gamma - \mathcal{G}(\infty)]\mathcal{S}\}^\dagger. \tag{100}$$

This relation holds generally. When $\mathcal{S}$ is invertible, we can insert Eq. (98) on the left hand side and right-multiply with $\mathcal{S}^{\dagger-1}$ to obtain the result Eq. (84):

$$i\Gamma\mathcal{I} - \mathcal{P}\overline{\mathcal{G}(\infty)}\mathcal{P} = [\mathcal{S}^{-1}\mathcal{G}(\infty)\mathcal{S}]^\dagger. \tag{101}$$

Note that $\mathcal{S}$ is invertible if and only if $\{\big(\hat{k}'_{j_i}(g_i(\infty))\big|\}$ is a linearly independent set, i.e., $\Pi^{(2)}(t)$ and $\Pi(t)$ have the same rank.

*Degeneracy of $g_i(\infty)$.*  In the main part we assume that $\mathcal{G}(\infty)$ and $\mathcal{K}(g_i(\infty))$ are diagonalizable, sharing nondegenerate stationary eigenvalues $g_i(\infty)$. However, the slip approximation can also be constructed if $\mathcal{G}(\infty)$ and $\hat{\mathcal{K}}(g_i(\infty))$ share an eigenvalue with the same degeneracy $d_i$. In the construction of the slip approximation the contributions of different eigenvectors $\big|g_i(\infty), l\big)$, $l = 1, \ldots, d_i$ to a degenerate eigenvalue $g_i(\infty)$ can be treated separately. However, when writing $\mathcal{S}$ as a sum of residuals $\operatorname{Res}\hat{\Pi}(g_i(\infty))$ in Eqs. (82a) and (83) we must count these independent contributions only once: the summation index $i$ must label the different eigenvalues of $\mathcal{G}(\infty)$, not the eigenvectors.

*Inconsistent approximations using $\mathcal{G}^{\mathrm{H}}(t)$ and $\mathcal{G}(t)^\dagger$.*  It is clear that the equation of motion $\frac{d}{dt}\Pi^{\mathrm{H}}(t) = i\Pi^{\mathrm{H}}(t)\mathcal{G}(t)^\dagger$, obtained by taking the adjoint of the time-local QME $\frac{d}{dt}\Pi(t) = -i\mathcal{G}(t)\Pi(t)$, leads to the semigroup approximation $\Pi^{(1)}(t)^\dagger = e^{i\mathcal{G}(\infty)^\dagger t}$ with initial slip correction $\Pi^{(2)}(t)^\dagger = \mathcal{S}^\dagger e^{i\mathcal{G}(\infty)^\dagger t}$. These approximations are equivalent to those discussed in the main text, but not related to the observable equation of motion.

*Semigroup approximation in Heisenberg picture.*  If one instead uses the generator $\mathcal{G}^{\mathrm{H}}$ [Eq. (47)] appearing in the *observable equation of motion*, $\frac{d}{dt}A(t) = i\mathcal{G}(t)^{\mathrm{H}}A(t)$ to construct

a semigroup approximation, one obtains a *different* result, $\Pi(t)^{\mathrm{H}} \approx e^{i\mathcal{G}^{\mathrm{H}}(\infty)t}$, which has several problems. It is not asymptotically exact and the required stationary generator $\lim_{t\to\infty}\mathcal{G}^{\mathrm{H}}(t)$ may fail to exist *even* when $\mathcal{G}(\infty)$ exists [Eq. (49)]. This problem can be circumvented by constructing a generator from the duality relation Eq. (84) as $\mathcal{G}^{\mathrm{H}}_{\mathrm{fix}}(\infty) = [\mathcal{S}^{-1}\mathcal{G}(\infty)\mathcal{S}]^{\dagger} = i\Gamma\mathcal{I} - \mathcal{P}\overline{\mathcal{G}(\infty)}\mathcal{P}$ where it is important that the long time limit $\lim_{t\to\infty}\mathcal{G}(t) = \mathcal{G}(\infty)$ is taken *before* inserting dual parameters, avoiding the trouble with $\lim_{t\to\infty}\bar{\mathcal{G}}(t)$. Using this "fixed" stationary generator for observables we then obtain a semigroup approximation $\mathcal{S}^{\dagger-1}e^{-i\mathcal{G}^{\mathrm{H}}_{\mathrm{fix}}(\infty)}\mathcal{S}^{\dagger}$ with slip approximation $e^{-i\mathcal{G}^{\mathrm{H}}_{\mathrm{fix}}(\infty)}\mathcal{S}^{\dagger}$ which coincide with the above mentioned adjoints of Schrödinger picture approximations $\Pi^{(1)}(t)^{\dagger}$ and $\Pi^{(2)}(t)^{\dagger}$, respectively.

## E  Fixed-point relation generator and memory kernel

Here we verify the consistency of the fermionic duality relations (49) and (67) applicable to a broad class of fermionic systems with the *general, exact* connection between the generator $\mathcal{G}(t)$ and the memory kernel $\mathcal{K}(t)$ established in Ref. [5]. This relation takes the form of a *functional fixed-point equation*:

$$\mathcal{G}(t) = \hat{\mathcal{K}}[\mathcal{G}](t) := \int_0^t ds\, \mathcal{K}(t-s)\mathcal{T}_{\to}e^{i\int_s^t dr\,\mathcal{G}(r)}, \tag{102}$$

with anti-timeordering $\mathcal{T}_{\to}$. In Ref. [5] it was shown that for $t \to \infty$ this gives the stationary fixed-point relation (78) used in the main text.

Analogous to Eq. (102) the Heisenberg generator $\mathcal{G}^{\mathrm{H}}(t) = [\Pi(t)^{-1}\mathcal{G}(t)\Pi(t)]^{\dagger}$ also obeys a functional fixed-point equation with the memory kernel $\mathcal{K}^{\mathrm{H}}(t) = \mathcal{K}(t)^{\dagger}$:

$$\mathcal{G}^{\mathrm{H}}(t) = \hat{\mathcal{K}}^{\mathrm{H}}[-\mathcal{G}^{\mathrm{H}}](t) := \int_0^t ds\, \mathcal{K}^{\mathrm{H}}(t-s)\mathcal{T}_{\to}e^{-i\int_s^t dr\,\mathcal{G}^{\mathrm{H}}(r)}. \tag{103}$$

The proof of this relation is analogous to the proof of the functional fixed-point equation in Schrödinger picture in Ref. [5]. In neither of the above general relations fermionic duality was used and it is thus important to check that it is consistent with these relations. To see this, substitute (49) and (67) in Eq. (103) to recover Eq. (102) evaluated at dual parameters:

$$\mathcal{G}^{\mathrm{H}}(t) = i\Gamma\mathcal{I} - \mathcal{P}\bar{\mathcal{G}}(t)\mathcal{P} \tag{104a}$$

$$\overset{!}{=} \int_0^t ds\left[i\Gamma\mathcal{I}\delta(t-s) - e^{-\Gamma(t-s)}\mathcal{P}\bar{\mathcal{K}}(t-s)\mathcal{P}\right]\mathcal{T}_{\to}e^{-i\int_s^t[i\Gamma\mathcal{I}-\mathcal{P}\bar{\mathcal{G}}(r)\mathcal{P}]dr} \tag{104b}$$

$$= i\Gamma\mathcal{I} - \mathcal{P}\int_0^t ds\,\bar{\mathcal{K}}(t-s)\mathcal{T}_{\to}e^{i\int_s^t dr\,\bar{\mathcal{G}}(r)}\mathcal{P}. \tag{104c}$$

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
