# Peer review of "Fermionic duality: General symmetry of open systems with strong dissipation and memory"

_SciPost Physics, doi:SciPost Phys. 11, 053 (2021)_

## Round 1 · Referee Report · Anonymous (Referee 1) · 2021-5-27

Strengths

1- Very clear explanation with plentiful of details.

2- Tackles a very interesting subject, providing a useful guideline to tackle nontrivial open system problems.

Weaknesses

1 - Sometimes difficult to follow .

Report

In this article, the authors detail a fermionic duality relation between the evolution of states and those of the observable. Such a relation constrain the dynamics and allows simplifying computations for a wide class of fermionic models.

I have very much enjoyed reading this paper, and I appreciated the effort the authors went into to make the paper accessible to a broad audience, by clearly stating the nature of the problem, how it can be understood, and why it is relevant. As such, I can commend its publication as it stands, although I have some suggestions (below) which the authors may want to consider.

1- A point which I find unclear is the recurring discussion of the non-physicality of the duality mapping. While I get that the map (e.g., $\bar \Pi(t)$) does not correspond to any physical system evolving, this can be seen as just a theoretical artifact and $\bar \Pi(t)$ never actually represent the physics of the system. On the contrary, this duality could be used to simulate "unphysical" systems using physical ones. By appropriately transforming the "physical" observables, one can simulate the non-physical domain. As such, I wonder if the authors have thought about this perspective. Obviously, this is just a suggestion, the article is clear as it is.

Other minor details: 2- The discussion about the conditions (I-III) in Section 3 is not very clear. In particular, it not clear in the following discussions how these three conditions ensure the fermionic duality relation. 3- It is a little confusing how Eq. (41) is derived from Eq. (40). Maybe the discussion about the sum rules can be slightly reformulated to make it more straightforward. 4- Concerning 4.2, I think it is closely related to the discussion in [Phys. Rev. Lett. 124, 190402] about the meaning of quantum jumps for divisible open quantum system dynamics. 5- Some similar "unphysical" modelling for bosonic-like systems have been discussed also in [ Phys. Rev. Lett. 120, 030402 (2018a),Nat Commun 10, 3721 (2019)]. I wonder if there is any connection with this work. 6-Very minor point: in the contourplot, sometimes the quantity plotted is indicated above the plot, some other times next to the colorbar, or it is in the caption. Although it is not a problem, it may be better to keep all the plot consistant. 7- Very minor point: Hermitian is often written as "hermitian". I think the correct spelling is Hermitian.

Requested changes

1 - Maybe comment on the point 1. 2- Clarify point 2. 3- Consider to comment on the other points raised in the report.

  • validity: high
  • significance: high
  • originality: high
  • clarity: good
  • formatting: excellent
  • grammar: good

Author:  Valentin Bruch  on 2021-06-25  [id 1526]

(in reply to Report 1 on 2021-05-27)
Category:
answer to question

We are glad to hear the referee enjoyed reading the manuscript and we are grateful for the interesting suggestions which have improved the paper.

A point which I find unclear is the recurring discussion of the non-physicality of the duality mapping. While I get that the map (e.g., $\bar{\Pi}(t)$) does not correspond to any physical system evolving, this can be seen as just a theoretical artifact and $\bar{\Pi}(t)$ never actually represent the physics of the system.

We are glad the referee brings this up. In the revised Introduction on p. 6 we now motivate this more explicitly from the beginning:

  1. This discussion recurs in the paper because in each of the discussed approaches the (lack of) CP is expressed by subtly different (failure to obey certain) restrictions. In some of the approaches - originally used to derive duality - it is impossible to see this. Also, as discussed under point 5, this precise notion of unphysical system is important to avoid unnecessary confusion with "effective" systems used in other works.
  2. Although in our own prior generally applicable work [10] it was clear that the - formally derived - inversion of the sign of the coupling rate $\Gamma$ was "unphysical" in some unspecified sense, the present work identifies this as the loss of complete positivity by employing the precise concepts of quantum information.
  3. This precision is important, since one of the central ideas the present work intends to convey is that the duality relation is a property of a class of systems (or type of Hamiltonians), not a specific one. This class is characterized by functional expressions for all its evolutions expressed in terms of model parameters. Of interest are not just a set of parameters, not even just the set of all possible physical parameter values, but the full domain including unphysical values. The unconventional insight is that the expressions at unphysical parameter values provide powerful analytic tools.
  4. This unphysicality distinguishes the present work from our prior work focusing on the weak-coupling limit where one can circumvent this as now explained in footnote 20. One can then obtain a physical dual system which brings very interesting and unconventional physical insights.

On the contrary, this duality could be used to simulate "unphysical" systems using physical ones. By appropriately transforming the "physical" observables, one can simulate the non-physical domain. As such, I wonder if the authors have thought about this perspective. Obviously, this is just a suggestion, the article is clear as it is.

  1. The duality indeed provides a way to describe the evolution of the dual systems using the original system. In the weak coupling limit this reverse use of the duality is indeed interesting and useful. In [17] it was shown for example, how repulsive physical systems have a physical attractive dual and vice versa. This was mentioned on p. 4 but is now pointed out more clearly in the revised manuscript.
  2. However, the focus of the present manuscript is on arbitrary coupling where in general no physical dual system can be introduced. The strong negative coupling rate induced by the environment through anti-Hermitian coupling prohibits this. As a result, every contribution to the dual evolution which changes the fermion parity on the system will lead to negative probabilities for the dual system if the initial state is entangled with some reference system (because $\bar{\Pi}(t)$ violates complete positivity). This uproots the statistical meaning of all resulting expressions (and, one may argue, the entire concepts of "system" and "evolution" which we nevertheless use for simplicity of presentation). Unless one has some specific mathematical interest in such type of expressions we cannot see how the reverse usage of the general duality can be of physical interest. The revised manuscript now points this out on p. 6.

The discussion about the conditions (I-III) in Section 3 is not very clear. In particular, it not clear in the following discussions how these three conditions ensure the fermionic duality relation.

We thank the referee for noting this point.

  1. We now further emphasize in the paper that these assumptions are at the basis of the quite involved derivation of the duality, which is not the subject of the present manuscript since it has been reported in our earlier work in great detail [10]. Importantly, this prior derivation does not lead to the insights presented in the present manuscript using quantum-information frameworks. The referee is correct that from the conditions (I-III) one cannot understand the results obtained in the present manuscript: One must start from the duality relation Eq. (3) derived earlier.
  2. We note that in forthcoming work (Bruch, Schulenborg, Wegewijs) a new and simpler derivation of the duality will be reported that can be applied within the various frameworks discussed in the present manuscript.

It is a little confusing how Eq. (41) is derived from Eq. (40). Maybe the discussion about the sum rules can be slightly reformulated to make it more straightforward.

We thank the referee for the careful reading, noting that our argument for deriving Eq. (41) was not sufficient, requiring the orthonormality of the measurement operators (not just their normalization). We corrected this.

Concerning 4.2, I think it is closely related to the discussion in [Phys. Rev. Lett. 124, 190402] about the meaning of quantum jumps for divisible open quantum system dynamics.

We thank the referee for pointing out this interesting reference. Indeed, different types of divisibility are at the basis of different stochastic simulation methods: whereas CP-divisibility of the dynamics enables an implementation that directly uses the jump operators to generate jumps, P-divisibility allows only for an indirect implementation where the jump operators must be constructed from Diosi's rate-operator. As the mentioned paper shows, any evolution - irrespective of whether it is divisible in some way or not - can be simulated by including reverse quantum jumps connecting trajectories. These 3 types of stochastic simulations are thus tied to 2 types of divisibility or lack thereof. However, both are causal.

By contrast, in our study we stumbled upon the new concept of anti-causal divisibility, which we show has a clear operational formulation as a causally reversed simulation task. For the resonant level model we found that this anti-causal CP-divisibility is lost while the dynamics remains causally CP-divisible. Anti-causal divisibility thus falls outside the scope of the distinctions between the existing stochastic simulation methods. It is an interesting open question how it impacts on the simulation trajectories.

In the revised paper, these questions are now raised briefly while citing the mentioned work. In summary we stress that

  1. fermionic duality surprisingly enables conclusions about anti-causal divisibility and
  2. this apparently new concept has a general operational formulation, independent of fermionic duality.

Some similar "unphysical" modelling for bosonic-like systems have been discussed also in [Phys. Rev. Lett. 120, 030402 (2018a), Nat Commun 10, 3721 (2019)]. I wonder if there is any connection with this work.

We thank the referee for bringing up the relation of these interesting works to the fermionic duality which we note appeared earlier [10]. The focus of our present paper is to extend this established result to the various quantum-information related frameworks. In a separate part of the Discussion (Section 6) of the revised manuscript we now include an abbreviated version of the following comparison which answers the referee's question.

Regarding [83] (Nat Commun 10, 3721) This work is indeed closely related to our prior work [10] at several key points:

  1. assumption of flat-band coupling
  2. "unphysicality" by a non-Hermitian coupling, referred to as "pseudo"
  3. proof by comparison of pair contractions corresponding to their two-time correlations functions (p. 5, following the method of the other reference [82] (Phys. Rev. Lett. 120, 030402))

There is however no simple relation as there are several key differences:

  1. their reservoirs are bosonic instead of fermionic
  2. they consider a specific system (spin-boson model), whereas duality was shown to apply to arbitrary local fermionic systems
  3. their modes with non-Hermitian coupling are added to the original modes with Hermitian coupling whereas in our dual model the coupling of all original modes is made anti-Hermitian without adding new modes. In this sense our "unphysicality" is more radical.
  4. they rely on numerical fitting of the low-temperature reservoir correlation functions which is accurate only for specific parameter range of interest (p. 4), whereas our duality was analytically derived by exploiting a general exact decomposition of the reservoir correlation function for any parameter set. (This decomposition is motivated by causality [13] and defines the renormalized perturbation expansion mentioned in Section 3 and 6 in the present manuscript.)
  5. they need a rotating-wave approximation which is not required for fermionic duality.

The only point that actually relates Ref. [83] to to our present manuscript is the observation that their "pseudo" model "does not guarantee complete positivity" (p. 5). Our result is stronger: by explicit construction we prove in Section 3.2.3 that the dual mode definitely breaks complete positivity whenever the original dynamics does not.

Regarding [82] (Phys. Rev. Lett. 120, 030402): This work provided the foundation for Ref. [83] and it is more loosely related to our prior work [10] and to the present manuscript:

  1. Their approach involves the construction of an effective, dissipatively evolving environment with less degrees of freedom which can be treated easier than the original reservoir. The local system is left untouched and the coupling remains physical. By contrast, duality changes the complete system - including the local Hamiltonian - to obtain a dual system of similar complexity as the original system: to exploit the functional parameter dependence of the dynamics one should "move within" the class of systems. In other words, their approach involves solving a problem from a different class of complexity, whereas the duality considers the same class of problems.
  2. The reduced evolution of the system is shown to be equal for their effective environment and for the original environment. The effective positive dissipative rates $\Gamma$ (in our notation) are converted into complex-imaginary amplitudes $t=i\sqrt{\Gamma}$ of a Hermitian coupling [Eqs. (3) and (11) loc. cit.]. This generates CP-TP dynamics which is certainly not unphysical - in the terminology of both us and Lambert et al - it is just "effective" (see our answer regarding "unphysicality" above). By contrast, in our work the entire coupling operator is made non-Hermitian (in fact, anti-Hermitian, by formally treating conjugate amplitudes independently, $ t \to i t$ but $t^\ast \to i t^{\ast}$ !), leading to the unphysical evolution of system + reservoirs. As a result, the coupling $\Gamma \propto t^\ast t \to -\Gamma$ becomes negative and the reduced dynamics becomes unphysical (non-CP).

Very minor point: in the contourplot, sometimes the quantity plotted is indicated above the plot, some other times next to the colorbar, or it is in the caption. Although it is not a problem, it may be better to keep all the plot consistant.

The definition of the quantity plotted is indicated next to the colorbar except in multipanel plots with a shared colorbar where this is not possible. This is preferred to avoid confusion.

Very minor point: Hermitian is often written as "hermitian". I think the correct spelling is Hermitian.

This has been fixed.

Attachment:

paper-changes.pdf

---

## Round 1 · Referee Report · Anonymous (Referee 2) · 2021-5-29

Strengths

1- Numerous details and various approaches illustrated 2- Presents useful method for simplifying solutions of open quantum dynamics

Weaknesses

1 - A bit long and at times difficult to follow 2- Unclear what is relation to previous work

Report

The article is very interesting and will be useful to both experts and people a bit further out of the field because it gives numerous details and reviews lots of previous results.

I think that the main usefulness of the new symmetry comes from the possibility to relate left and right eigenvectors via conjugation and parity transformation and that there are cross-relations in the spectrum. The paper goes into the implication of this transformation for various kinds of quantum dynamics beyond the Lindblad equation.

My main point that I would like the authors to address is to better contrast their work with previous work. More specifically, the Lindblad (Markovian) PT symmetry (Ref. [15,16]). Is the main difference that they work with more general quantum map than the ones generated by the Lindblad master equation? The symmetry they introduce seems also a bit similar to weak symmetries in Lindblad master equations: New J. Phys. 14 073007 (2012).

Minor points:

1 - In figure 5 the authors compare the initial slip approximation with the nonperturbative semigroup for a resonant level model. It would be good to have an explanation in the caption how their novel results come into play in this study.

2 - I would have liked to have seen more examples of their duality being used to help compute physical quantities. I leave it as an optional suggestion to the the authors to include another physical example.

Requested changes

1 - Compare with previous work in more detail 2- Explain caption of figure 5 in relation to their work

  • validity: top
  • significance: high
  • originality: good
  • clarity: good
  • formatting: good
  • grammar: good

Author:  Valentin Bruch  on 2021-06-25  [id 1527]

(in reply to Report 2 on 2021-05-29)
Category:
answer to question

We are glad the referee finds the article very interesting and useful and we are grateful for the helpful feedback.

My main point that I would like the authors to address is to better contrast their work with previous work.

In the revised manuscript we have devoted to this a separate part of the Discussion (Section 6). We also added more references and clarified the relation of duality to these approaches.

More specifically, the Lindblad (Markovian) PT symmetry (Ref. [15,16]). Is the main difference that they work with more general quantum map than the ones generated by the Lindblad master equation?

This is correct. This is now pointed out when this reference is cited in Section 6.

The symmetry they introduce seems also a bit similar to weak symmetries in Lindblad master equations: New J. Phys. 14 073007 (2012).

We discuss this interesting reference at the end (Section 6) of the revised manuscript to clarify in what (loose) sense fermionic duality is a "symmetry" as the title - for lack of better words - suggests.

  1. Weak symmetry is defined in [81] (New J. Phys. 14 073007) as the time-constant generator commuting with a symmetry superoperator, which in our case of general time-local generators can be generalized to $[\mathcal{S},\mathcal{G}(t)]=0$. This applies to the fermion-parity unitary conjugation $\mathcal{S} = (-1)^N \bullet (-1)^N$. Although this weak symmetry property is used in the original derivation of duality [10] and in the present analysis (App. C) this is not the same thing as fermionic duality. In particular, note that this $\mathcal{S}$ is not the superoperator $\mathcal{P}=(-1)^N \bullet$ of parity left-multiplication which appears in all our duality relations and which is not a unitary conjugation. (The same holds for right multiplication which gives an equivalent formulation of duality.)
  2. Strong symmetry is defined in [81] as the Hamiltonian and jump operators commuting with a unitary symmetry operator $S$ which can be generalized to the time-local case $[S,L_\alpha(t)]=[S,H(t)]=0$. In our case there is no such strong symmetry, but something very close to it which we exploit in our present analysis: For the unitary parity operator $S=(-1)^N$ we have $[(-1)^N,H(t)]=0$ and the jump operators do have definite parity, $(-1)^N L_\alpha (-1)^N = \pm L_\alpha$, but no fixed sign. In general, either sign of the parity can occur, see 4.2.1 and App. C.

Minor points: In figure 5 the authors compare the initial slip approximation with the nonperturbative semigroup for a resonant level model. It would be good to have an explanation in the caption how their novel results come into play in this study.

We have revised the caption pointing out that the apparent success of the initial slip approximation in Fig. 5(a) makes its complete failure in 5(b) unexpected. The novel insight is that fermionic duality dictates by Eq. (86) that this failure occurs at the isolated parameter points in Eq. (87). The formulation of the initial slip itself using exact quantities, Eq. (80), is based on the recent result reported by the authors in Ref. [5].

I would have liked to have seen more examples of their duality being used to help compute physical quantities. I leave it as an optional suggestion to the the authors to include another physical example.

  1. Various ways in which duality can be exploited can be seen already clearly by the weak-coupling applications in previous works cited on p. 4 of the introduction.
  2. Strong coupling problems, although of higher interest, do not fundamentally changes this but mostly add complications of technical nature. In fact, basic implications of fermionic duality were discovered and partially exploited in the renormalization group study [13], illustrating its use in a very technical context. This is now mentioned in Section 6.
  3. Since we share the referees concern that the paper is "a bit long" we see no possibility to properly explain the application to another, complicated model.
  4. In the revised discussion it is also stressed that the results of Sections 1-4 and most of the conclusions in the application Section 5 hold not only for a single model but for the entire class of models for which fermionic duality applies (as defined in Sec. 3 and references cited there). These include strongly interacting non-equilibrium quantum transport models such as the above mentioned Anderson model. In the present manuscript the noninteracting resonant level model was primarily discussed because it allows to analytically illustrate key features of duality without unnecessary complications by interactions. In particular, for interacting models the application discussion in Section 5 goes through, but for these the analytic calculation of the required evolution quantities - the prerequisite for fermionic duality analysis - is of course very challenging and outside the scope of this paper. The merit of duality lies in simplifying a given calculation given a method of choice, not in providing this method.

Attachment:

paper-changes_c2esxIf.pdf

---

## Round 2 · Referee Report · Anonymous (Referee 2) · 2021-8-26

Report

The authors have addressed my remarks and I recommend publication.

---

## Round 2 · List of Changes

• The explanation of the relation to other works has been extended and moved to a dedicated paragraph in Section 6 (p. 35f.).
  • page 4: The explanation of the special case of fermionic duality in the weak coupling limit has been clarified.
  • page 6: The discussion of the unphysical properties of the dual propagator has been extended. It is now mentioned that only the unphysical dual system can lead to the unconventional insights of fermionic duality.
  • page 12: It is now clarified that the assumptions (I)-(III) are required to derive the duality relation, but are not discussed in this work.
  • page 17: The explanation of Eq. (41) has been corrected. It requires the orthonormality of the canonical measurement operators.
  • page 27: Footnote 20 has been added and provides a connection to the weak coupling duality.
  • page 32: The caption of Figure 5 has been rewritten to clarify the role of fermionic duality in this figure.
  • page 35f.: Section 6 has be restructured. It now contains a summary, the relation to other works and the outlook as separate parts. The goal of the duality ("simplifying a calculation given a method of choice") is now stated more explicitly.
  • page 36: The outlook has been rewritten and adapted to the new structure of Section 6.

---

## Editorial Decision

published